# Accelerated Training via Incrementally Growing Neural Networks using Variance Transfer and Learning Rate Adaptation

**Xin Yuan**
University of Chicago
yuanx@uchicago.edu

**Pedro Savarese**
TTI-Chicago
savarese@ttic.edu

**Michael Maire**
University of Chicago
mmaire@uchicago.edu

## Abstract

We develop an approach to efficiently grow neural networks, within which parameterization and optimization strategies are designed by considering their effects on the training dynamics. Unlike existing growing methods, which follow simple replication heuristics or utilize auxiliary gradient-based local optimization, we craft a parameterization scheme which dynamically stabilizes weight, activation, and gradient scaling as the architecture evolves, and maintains the inference functionality of the network. To address the optimization difficulty resulting from imbalanced training effort distributed to subnetworks fading in at different growth phases, we propose a learning rate adaption mechanism that rebalances the gradient contribution of these separate subcomponents. Experiments show that our method achieves comparable or better accuracy than training large fixed-size models, while saving a substantial portion of the original training computation budget. We demonstrate that these gains translate into real wall-clock training speedups.

## 1 Introduction

Modern neural network design typically follows a "larger is better" rule of thumb, with models consisting of millions of parameters achieving impressive generalization performance across many tasks, including image classification [22, 32, 30, 46], object detection [13, 26, 11], semantic segmentation [27, 3, 24] and machine translation [34, 7]. Within a class of network architecture, deeper or wider variants of a base model typically yield further improvements to accuracy. Residual networks (ResNets) [15] and wide residual networks [45] illustrate this trend in convolutional neural network (CNN) architectures. Dramatically scaling up network size into the billions of parameter regime has recently revolutionized transformer-based language modeling [34, 7, 1].

The size of these models imposes prohibitive training costs and motivates techniques that offer cheaper alternatives to select and deploy networks. For example, hyperparameter tuning is notoriously expensive as it commonly relies on training the network multiple times, and recent techniques aim to circumvent this by making hyperparameters transferable between models of different sizes, allowing them to be tuned on a small network prior to training an original large model once [41].

Our approach incorporates these ideas, but extends the scope of transferability to include the parameters of the model itself. Rather than view training small and large models as separate events, we grow a small model into a large one through many intermediate steps, each of which introduces additional parameters to the network. Our contribution is to do so in a manner that preserves the function computed by the model at each growth step (functional continuity) and offers stable training dynamics, while also saving compute by leveraging intermediate solutions. More specifically, we use partially trained subnetworks as scaffolding that accelerates training of newly added parameters, yielding greater overall efficiency than training a large static model from scratch.

37th Conference on Neural Information Processing Systems (NeurIPS 2023).

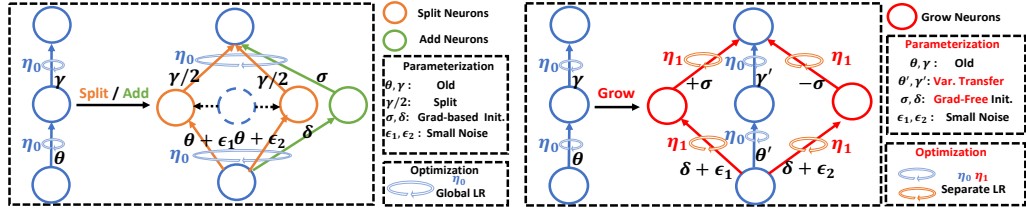

(a) Existing Methods: Splitting Init, Global LR    (b) Ours: Function-Preserving Init, Stagewise LR

Figure 1: Dynamic network growth strategies. Different from (a) which rely on either splitting [4, 25, 39] or adding neurons with auxiliary local optimization [38, 10], our initialization (b) of new neurons is random but function-preserving. Additionally, our separate learning rate scheduler governs weight updating to address the discrepancy in total accumulated training between different growth stages.

Competing recent efforts to grow deep models from simple architectures [4, 23, 5, 25, 39, 37, 38, 44, 10] draw inspiration from other sources, such as the progressive development processes of biological brains. In particular, Net2Net [4] grows the network by randomly splitting learned neurons from previous phases. This replication scheme, shown in Figure 1(a) is a common paradigm for most existing methods. Gradient-based methods [38, 39] determine which neurons to split and how to split them by solving a combinatorial optimization problem with auxiliary variables.

At each growth step, naive random initialization of new weights destroys network functionality and may overwhelm any training progress. Weight rescaling with a static constant from a previous step is not guaranteed to be maintained as the network architecture evolves. Gradient-based methods outperform these simple heuristics but require additional training effort in their parameterization schemes. Furthermore, all existing methods use a global LR scheduler to govern weight updates, ignoring the discrepancy among subnetworks introduced in different growth phases. The gradient itself and other parameterization choices may influence the correct design for scaling weight updates.

We develop a growing framework around the principles of enforcing transferability of parameter settings from smaller to larger models (extending [41]), offering functional continuity, smoothing optimization dynamics, and rebalancing learning rates between older and newer subnetworks. Figure 1(b) illustrates key differences with prior work. Our core contributions are:

- **Parameterization using Variance Transfer:** We propose a parameterization scheme accounting for the variance transition among networks of smaller and larger width in a single training process. Initialization of new weights is gradient-free and requires neither additional memory nor training.
- **Improved Optimization with Learning Rate Adaptation:** Subnetworks trained for different lengths have distinct learning rate schedules, with dynamic relative scaling driven by weight norm statistics.
- **Better Performance and Broad Applicability:** Our method not only trains networks fast, but also yields excellent generalization accuracy, even outperforming the original fixed-size models. Flexibility in designing a network growth curve allows choosing different trade-offs between training resources and accuracy. Furthermore, adopting an adaptive batch size schedule provides acceleration in terms of wall-clock training time. We demonstrate results on image classification and machine translation tasks, across various network architectures.

## 2   Related Work

**Network Growing.** A diverse range of techniques train models by progressively expanding the network architecture [36, 9, 5, 37, 44]. Within this space, the methods of [4, 25, 39, 38, 10] are most relevant to our focus – incrementally growing network width across multiple training stages. Net2Net [4] proposes a gradient-free neuron splitting scheme via replication, enabling knowledge transfer from previous training phases; initialization of new weights follows simple heuristics. Liu *et al.*'s splitting approach [25] derives a gradient-based scheme for duplicating neurons by formulating a combinatorial optimization problem. FireFly [38] gains flexibility by also incorporating brand new neurons. Both methods improve Net2Net's initialization scheme by solving an optimization problem with auxiliary variables, at the cost of extra training effort. GradMax [10], in consideration of training dynamics, performs initialization via solving a singular value decomposition (SVD) problem.

**Neural Architecture Search (NAS) and Pruning.** Another subset of methods mix growth with dynamic reconfiguration aimed at discovering or pruning task-optimized architectures. Network Morphism [36] searches for efficient networks by extending layers while preserving the parameters. AutoGrow [37] takes an AutoML approach governed by heuristic growing and stopping policies. Yuan *et al.* [44] combine learned pruning with a sampling strategy that dynamically increases or decreases network size. Unlike these methods, we focus on the mechanics of growth when the target architecture is known, addressing the question of how to best transition weight and optimizer state to continue training an incrementally larger model. NAS and pruning are orthogonal to, though potentially compatible with, the technical approach we develop.

**Hyperparameter Transfer.** Multiple works [42, 29, 16] explore transferable hyperparameter (HP) tuning. The recent Tensor Program (TP) work of [40] and [41] focuses on zero-shot HP transfer across model scale and establishes a principled network parameterization scheme to facilitate HP transfer. This serves as an anchor for our strategy, though, as Section 3 details, modifications are required to account for dynamic growth.

**Learning Rate Adaptation.** Surprisingly, the existing spectrum of network growing techniques utilize relatively standard learning rate schedules and do not address potential discrepancy among subcomponents added at different phases. While general-purpose adaptive optimizers, *e.g.*, Ada-Grad [8], RMSProp [33], Adam [20], or AvaGrad [31], might ameliorate this issue, we choose to explicitly account for the discrepancy. As layer-adaptive learning rates (LARS) [12, 43] benefit in some contexts, we explore further learning rate adaption specific to both layer and growth stage.

## 3 Method

### 3.1 Parameterization and Optimization with Growing Dynamics

**Functionality Preservation**. We grow a network's capacity by expanding the width of computational units (*e.g.,* hidden dimensions in linear layers, filters in convolutional layers). To illustrate our scheme, consider a 3-layer fully-connected network with ReLU activations $\phi$ at a growing stage $t$:

$$\boldsymbol{u}_t = \phi(\boldsymbol{W}_t^x \boldsymbol{x}) \qquad \boldsymbol{h}_t = \phi(\boldsymbol{W}_t^u \boldsymbol{u}_t) \qquad \boldsymbol{y}_t = \boldsymbol{W}_t^h \boldsymbol{h}_t \,, \tag{1}$$

where $\boldsymbol{x} \in \mathbb{R}^{C^x}$ is the network input, $\boldsymbol{y}_t \in \mathbb{R}^{C^y}$ is the output, and $\boldsymbol{u}_t \in \mathbb{R}^{C_t^u}, \boldsymbol{h}_t \in \mathbb{R}^{C_t^h}$ are the hidden activations. In this case, $\boldsymbol{W}_t^x$ is a $C_t^u \times C^x$ matrix, while $\boldsymbol{W}_t^u$ is $C_t^h \times C_t^u$ and $\boldsymbol{W}_t^h$ is $C^y \times C_t^h$. Our growing process operates by increasing the dimensionality of each hidden state, *i.e.,* from $C_t^u$ and $C_t^h$ to $C_{t+1}^u$ and $C_{t+1}^h$, effectively expanding the size of the parameter tensors for the next growing stage $t + 1$. The layer parameter matrices $\boldsymbol{W}_t$ have their shapes changed accordingly and become $\boldsymbol{W}_{t+1}$. Figure 2 illustrates the process for initializing $\boldsymbol{W}_{t+1}$ from $\boldsymbol{W}_t$ at a growing step.[1]

Following Figure 2(a), we first expand $\boldsymbol{W}_t^x$ along the output dimension by adding two copies of new weights $\boldsymbol{V}_t^x$ of shape $\frac{C_{t+1}^u - C_t^u}{2} \times C^x$, generating new features $\phi(\boldsymbol{V}_t^x \boldsymbol{x})$. The first set of activations become

$$\boldsymbol{u}_{t+1} = \text{concat}\left(\boldsymbol{u}_t, \phi(\boldsymbol{V}_t^x \boldsymbol{x}), \phi(\boldsymbol{V}_t^x \boldsymbol{x})\right) \,, \tag{2}$$

where $\text{concat}$ denotes the concatenation operation. Next, we expand $\boldsymbol{W}_t^u$ across both input and output dimensions, as shown in Figure 2(b). We initialize new weights $\boldsymbol{Z}_t^u$ of shape $C_t^h \times \frac{C_{t+1}^u - C_t^u}{2}$ and add to $\boldsymbol{W}_t^u$ two copies of it with different signs: $+\boldsymbol{Z}_t^u$ and $-\boldsymbol{Z}_t^u$. This preserves the output of the layer since $\phi(\boldsymbol{W}_t^u \boldsymbol{u}_t + \boldsymbol{Z}_t^u \phi(\boldsymbol{V}_t^x \boldsymbol{x}) + (-\boldsymbol{Z}_t^u)\phi(\boldsymbol{V}_t^x \boldsymbol{x})) = \phi(\boldsymbol{W}_t^u \boldsymbol{u}_t) = \boldsymbol{h}_t$. We then add two copies of new weights $\boldsymbol{V}_t^u$, which has shape $\frac{C_{t+1}^h - C_t^h}{2} \times C_{t+1}^u$, yielding activations

$$\boldsymbol{h}_{t+1} = \text{concat}(\boldsymbol{h}_t, \phi(\boldsymbol{V}_t^u \boldsymbol{u}_{t+1}), \phi(\boldsymbol{V}_t^u \boldsymbol{u}_{t+1})) \,. \tag{3}$$

We similarity expand $\boldsymbol{W}_t^h$ with new weights $\boldsymbol{Z}_t^h$ to match the dimension of $\boldsymbol{h}_{t+1}$, as shown in Figure 2(c). The network's output after the growing step is:

$$\begin{aligned} \boldsymbol{y}_{t+1} &= \boldsymbol{W}_t^h \boldsymbol{h}_t + \boldsymbol{Z}_t^h \phi(\boldsymbol{V}_t^u \boldsymbol{u}_{t+1}) + (-\boldsymbol{Z}_t^h)\phi(\boldsymbol{V}_t^u \boldsymbol{u}_{t+1}) \\ &= \boldsymbol{W}_t^h \boldsymbol{h}_t = \boldsymbol{y}_t \,, \end{aligned} \tag{4}$$

which preserves the original output features in Eq. 1. Appendix B provides illustrations for more layers.

---

[1] We defer the transformation between $\boldsymbol{W}_t$ and $\boldsymbol{W}_t^{'}$ to the next subsection. It involves rescaling by constant factors, does not affect network functionality, and is omitted in Eq. 1- 4 for simplicity.

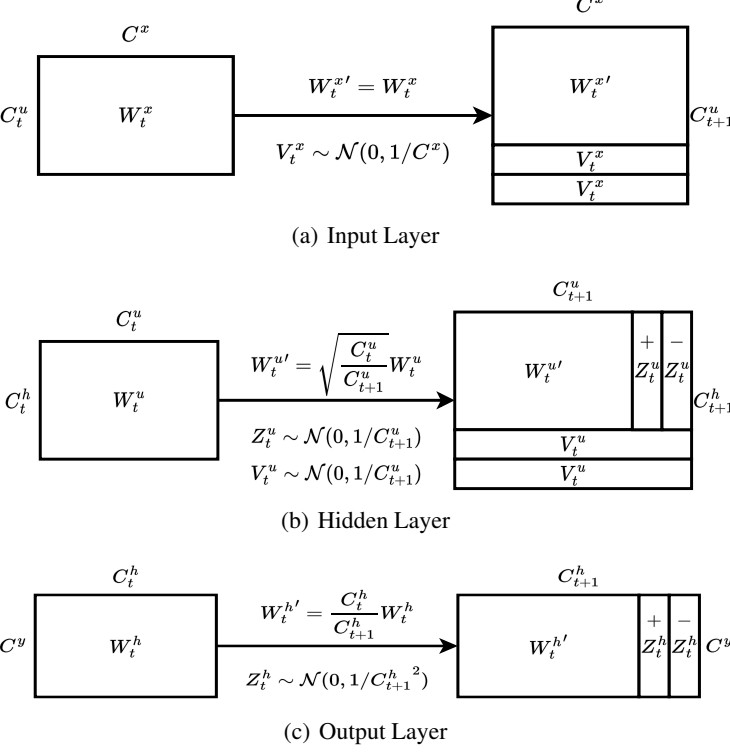

(a) Input Layer

(b) Hidden Layer

(c) Output Layer

Figure 2: Initialization scheme. In practice, we also add noise to the expanded parameter sets for symmetry breaking.

**Weights Initialization with Variance Transfer (VT).** Yang *et al.* [41] investigate weight scaling with width at initialization, allowing hyperparameter transfer by calibrating variance across model size. They modify the variance of output layer weights from the commonly used $\frac{1}{\text{fan}_{in}}$ to $\frac{1}{\text{fan}_{in}^2}$. We adopt this same correction for the added weights with new width: $\boldsymbol{W}^h$ and $\boldsymbol{Z}^h$ are initialized with variances of $\frac{1}{C_t^{h2}}$ and $\frac{1}{C_{t+1}^{h}{}^2}$.

However, this correction considers training differently-sized models separately, which fails to accommodate the training dynamics in which width grows incrementally. Assuming that the weights of the old subnetwork follow $\boldsymbol{W}_t^h \sim \mathcal{N}(0, \frac{1}{C_t^{h2}})$ (which holds at initialization), we make them compatible with new weight tensor parameterization by rescaling it with the fan$_{in}$ ratio as $\boldsymbol{W}_t^{h'} = \boldsymbol{W}_t^h \cdot \frac{C_t^h}{C_{t+1}^h}$. See Table 1 (top). Appendix A provides detailed analysis.

This parameterization rule transfers to modern CNNs with batch normalization (BN). Given a weight scaling ratio of $c$, the running mean $\mu$ and variance $\sigma$ of BN layers are modified as $c\mu$ and $c^2\sigma$, respectively.

**Stage-wise Learning Rate Adaptation (LRA).** Following [41], we employ a learning rate scaling factor of $\propto \frac{1}{\text{fan}_{in}}$ on the output layer when using SGD, compensating for the initialization scheme. However, subnetworks from different growth stages still share a global learning rate, though they have trained for different lengths. This may cause divergent behavior among the corresponding weights, making the training iterations after growing sensitive to the scale of the newly-initialized weights. Instead of adjusting newly added parameters via local optimization [38, 10], we govern the update of each subnetwork in a stage-wise manner.

Let $\mathcal{W}_t$ denote the parameter variables of a layer at a growth stage $t$, where we let $\boldsymbol{W}_t$ and $\boldsymbol{W}_t'$ correspond to the same set of variables such that $\mathcal{W}_{t+1} \setminus \mathcal{W}_t$ denotes the new parameter variables whose values are initialized with $\boldsymbol{Z}_t$ and $\boldsymbol{V}_t$. Moreover, let $\boldsymbol{W}_{\triangle k}$ and $\boldsymbol{G}_{\triangle k}$ denote the values and gradients of $\mathcal{W}_k \setminus \mathcal{W}_{k-1}$. We adapt the learning rate used to update each sub-weight $\boldsymbol{W}_{\triangle k}$, for

Table 1: Parameterization and optimization transition for different layers during growing. $C_t$ and $C_{t+1}$ denote the input dimension before and after a growth step.

| | | Input Layer | Hidden Layer | Output Layer |
|---|---|---|---|---|
| Init. | Old Re-scaling | 1 | $\sqrt{C_t^u/C_{t+1}^u}$ | $C_t^h/C_{t+1}^h$ |
| | New Init. | $1/C_t^x$ | $1/C_{t+1}^u$ | $1/(C_{t+1}^h)^2$ |
| Adapt. | 0-th Stage | 1 | 1 | $1/C_0$ |
| | $t$-th Stage | $\frac{\|\boldsymbol{W}_{\Delta t}^x\|}{\|\boldsymbol{W}_{\Delta 0}^x\|}$ | $\frac{\|\boldsymbol{W}_{\Delta t}^u\|}{\|\boldsymbol{W}_{\Delta 0}^u\|}$ | $\frac{\|\boldsymbol{W}_{\Delta t}^h\|}{\|\boldsymbol{W}_{\Delta 0}^h\|}$ |

$0 \leq k \leq t$, as follows:

$$\eta_k = \eta_0 \cdot \frac{f(\boldsymbol{W}_{\Delta k})}{f(\boldsymbol{W}_{\Delta 0})} \quad \boldsymbol{W}_{\Delta k} \leftarrow \boldsymbol{W}_{\Delta k} - \eta_k \boldsymbol{G}_{\Delta k}, \tag{5}$$

where $\eta_0$ is the base learning rate, $f$ is a function that maps subnetworks of different stages to corresponding train-time statistics, and $\boldsymbol{W}_{\Delta 0}$ are the layer's parameter variables at the first growth stage. Table 1 (bottom) summarizes our LR adaptation rule for SGD when $f$ is instantiated as weight norm, providing an stage-wise extension to the layer-wise adaptation method LARS [12], *i.e.*, $LR \propto \|\boldsymbol{W}\|$. Alternative heuristics are possible; see Appendices C and D.

### 3.2 Flexible and Efficient Growth Scheduler

We train the model for $T_{total}$ epochs by expanding the channel number of each layer to $C_{final}$ across $N$ growth phases. Existing methods [25, 38] fail to derive a systemic way for distributing training resources across a growth trajectory. Toward maximizing efficiency, we experiment with a coupling between model size and training epoch allocation.

**Architectural Scheduler.** We denote initial channel width as $C_0$ and expand exponentially:

$$C_t = \begin{cases} C_{t-1} + \lfloor p_c C_{t-1} \rceil_2 & \text{if} \quad t < N-1 \\ C_{final} & \text{if} \quad t = N-1 \end{cases} \tag{6}$$

where $\lfloor \cdot \rceil_2$ rounds to the nearest even number and $p_c$ is the growth rate between stages.

**Epoch Scheduler.** We denote number of epochs assigned to $t$-th training stage as $T_t$, with $\sum_{t=0}^{N-1} T_t = T_{total}$. We similarly adapt $T_t$ via an exponential growing scheduler:

$$T_t = \begin{cases} T_{t-1} + \lfloor p_t T_{t-1} \rceil & \text{if} \quad t < N-1 \\ T_{total} - \sum_{i=0}^{N-2} T_i & \text{if} \quad t = N-1 \end{cases} \tag{7}$$

---

**Algorithm 1** : Growing using Var. Transfer and Learning Rate Adapt. with Flexible Scheduler

**Input:** Data $\boldsymbol{X}$, labels $\boldsymbol{Y}$, task loss $L$
**Output:** Grown model $\mathcal{W}$
Initialize: $\mathcal{W}_0$ with $C_0, T_0, B_0, \eta_0$
**for** t = 0 **to** $N-1$ **do**
  **if** $t > 0$ **then**
    Init. $S_n$ from $S_{n-1}$ using VT in Table 1.
    Update $C_t$ and $T_t$ using Eq. 6 and Eq. 7.
    Update $B_t$ using Eq. 8 (optional)
    $\text{Iter}_{total} = T_t * len(X)//B_t$
  **end if**
  **for** iter = 1 **to** $\text{Iter}_{total}$ **do**
    Forward and calculate $l = L(\mathcal{W}_t(\boldsymbol{x}), \boldsymbol{y})$.
    Back propagation with $l$.
    Update each sub-component using Eq. 5.
  **end for**
**end for**
return $\mathcal{W}_{N-1}$

---

**Wall-clock Speedup via Batch Size Adaptation.** Though the smaller architectures in early growth stages require fewer FLOPs, hardware capabilities may still restrict practical gains. When growing width, in order to ensure that small models fully utilize the benefits of GPU parallelism, we adapt the batch size along with the exponentially-growing architecture in a reverse order:

$$B_{t-1} = \begin{cases} B_{base} & \text{if} \quad t = N \\ B_t + \lfloor p_b B_t \rceil & \text{if} \quad t < N \end{cases} \tag{8}$$

where $B_{base}$ is the batch size of the large baseline model. Algorithm 1 summarizes our full method.

# 4 Experiments

We evaluate on image classification and machine translation tasks. For image classification, we use CIFAR-10 [21], CIFAR-100 [21] and ImageNet [6]. For the neural machine translation, we use the IWSLT'14 dataset [2] and report the BLEU [28] score on German to English (De-En) translation.

**Large Baselines via Fixed-size Training.** We use VGG-11 [32] with BatchNorm [19], ResNet-20 [15], MobileNetV1 [17] for CIFAR-10 and VGG-19 with BatchNorm, ResNet-18, MobileNetV1 for CIFAR-100. We follow [18] for data augmentation and processing, adopting random shifts/mirroring and channel-wise normalization. CIFAR-10 and CIFAR-100 models are trained for 160 and 200 epochs respectively, with a batch size of 128 and initial learning rate (LR) of 0.1 using SGD. We adopt a cosine LR schedule and set the weights decay and momentum as 5e-4 and 0.9. For ImageNet, we train the baseline ResNet-50 and MobileNetV1 [17] using SGD with batch sizes of 256 and 512, respectively. We adopt the same data augmentation scheme as [14], the cosine LR scheduler with initial LR of 0.1, weight decay of 1e-4 and momentum of 0.9.

For IWSLT'14, we train an Encoder-Decoder Transformer (6 attention blocks each) [34]. We set width as $d_{model} = 1/4d_{ffn} = 512$, the number of heads $n_{head} = 8$ and $d_k = d_q = d_v = d_{model}/n_{head} = 64$. We train the model using Adam for 20 epochs with learning rate 1e-3 and $(\beta_1, \beta_2) = (0.9, 0.98)$. Batch size is 1500 and we use 4000 warm-up iterations.

Table 2: Growing ResNet-20, VGG-11, and MobileNetV1 on CIFAR-10.

| Method | ResNet-20 | | VGG-11 | | MobileNetv1 | |
|---|---|---|---|---|---|---|
| | Train Cost(%) ↓ | Test Accuracy(%) ↑ | Train Cost(%) ↓ | Test Accuracy(%) ↑ | Train Cost(%) ↓ | Test Accuracy(%) ↑ |
| Large Baseline | 100 | $92.62 \pm 0.15$ | 100 | $92.14 \pm 0.22$ | 100 | $92.27 \pm 0.11$ |
| Net2Net | **54.90** | $91.60 \pm 0.21$ | **52.91** | $91.78 \pm 0.27$ | **53.80** | $90.34 \pm 0.20$ |
| Splitting | 70.69 | $91.80 \pm 0.10$ | 63.76 | $91.88 \pm 0.15$ | 65.92 | $91.50 \pm 0.06$ |
| FireFly-split | 58.47 | $91.78 \pm 0.11$ | 56.18 | $91.91 \pm 0.15$ | 56.37 | $91.56 \pm 0.06$ |
| FireFly | 68.96 | $92.10 \pm 0.13$ | 60.24 | $92.08 \pm 0.16$ | 62.12 | $91.69 \pm 0.07$ |
| Ours | **54.90** | $\mathbf{92.53 \pm 0.11}$ | **52.91** | $\mathbf{92.34 \pm 0.15}$ | **53.80** | $\mathbf{92.01 \pm 0.10}$ |

Table 3: Growing ResNet-18, VGG-19, and MobileNetV1 on CIFAR-100.

| Method | ResNet-18 | | VGG-19 | | MobileNetv1 | |
|---|---|---|---|---|---|---|
| | Train Cost(%) ↓ | Test Accuracy(%) ↑ | Train Cost(%) ↓ | Test Accuracy(%) ↑ | Train Cost(%) ↓ | Test Accuracy(%) ↑ |
| Large Baseline | 100 | $78.36 \pm 0.12$ | 100 | $72.59 \pm 0.23$ | 100 | $72.13 \pm 0.13$ |
| Net2Net | **52.63** | $76.48 \pm 0.20$ | **52.08** | $71.88 \pm 0.24$ | **52.90** | $70.01 \pm 0.20$ |
| Splitting | 68.01 | $77.01 \pm 0.12$ | 60.12 | $71.96 \pm 0.12$ | 58.39 | $70.45 \pm 0.10$ |
| FireFly-split | 56.11 | $77.22 \pm 0.11$ | 54.64 | $72.19 \pm 0.14$ | 54.36 | $70.69 \pm 0.11$ |
| FireFly | 65.77 | $77.25 \pm 0.12$ | 57.48 | $72.79 \pm 0.13$ | 56.49 | $70.99 \pm 0.10$ |
| Ours | **52.63** | $\mathbf{78.12 \pm 0.15}$ | **52.08** | $\mathbf{73.26 \pm 0.14}$ | **52.90** | $\mathbf{71.53 \pm 0.13}$ |

**Implementation Details.** We compare with the growing methods Net2Net [4], Splitting [25], FireFly-split, FireFly [38] and GradMax [10]. In our method, noise for symmetry breaking is 0.001 to the norm of the initialization. We re-initialize the momentum buffer at each growing step when using SGD while preserving it for adaptive optimizers (*e.g.,* Adam, AvaGrad).

For image classification, we run the comparison methods except GradMax alongside our algorithm for all architectures under the same growing scheduler. For the architecture scheduler, we set $p_c$ as 0.2 and $C_0$ as 1/4 of large baselines in Eq. 6 for all layers and grow the seed architecture within $N = 9$ stages towards the large ones. For epoch scheduler, we set $p_t$ as 0.2, $T_0$ as 8, 10, and 4 in Eq. 7 on CIAFR-10, CIFAR-100, and ImageNet respectively. Total training epochs $T_{total}$ are the same as the respective large fixed-size models. We train the models and report the results averaging over 3 random seeds.

For machine translation, we grow the encoder and decoder layers' widths while fixing the embedding layer dimension for a consistent positional encoding table. The total number of growing stages is 4, each trained for 5 epochs. The initial width is 1/8 of the large baseline (*i.e.,* $d_{model} = 64$ and $d_{k,q,v} = 8$). We set the growing ratio $p_c$ as 1.0 so that $d_{model}$ evolves as 64, 128, 256 and 512.

We train all the models on an NVIDIA 2080Ti 11GB GPU for CIFAR-10, CIFAR-100, and IWSLT'14, and two NVIDIA A40 48GB GPUs for ImageNet.

## 4.1 CIFAR Results

All models grow from a small seed architecture to the full-sized one in 9 stages, each trained for $\{8, 9, 11, 13, 16, 19, 23, 28, 33\}$ epochs (160 total) on CIFAR-10, and $\{10, 12, 14, 17, 20, 24, 29, 35, 39\}$ (200 total) on CIFAR-100. Net2Net follows the design of growing by splitting via simple neuron replication, hence achieving the same training efficiency as our gradient-free method under the same growing schedule. Splitting and Firely require additional training effort for their neuron selection schemes and allocate extra GPU memory for auxiliary variables during the local optimization stage. This is computationally expensive, especially when growing a large model.

**ResNet-20, VGG-11, and MobileNetV1 on CIFAR-10.** Table 2 shows results in terms of test accuracy and training cost calculated based on overall FLOPs. For ResNet-20, Splitting and Firefly achieve better test accuracy than Net2Net, which suggests the additional optimization benefits neuron selection at the cost of training efficiency. Our method requires only $54.9\%$ of the baseline training cost and outperforms all competing methods, while yielding only $0.09p.p$ (percentage points) performance degradation compared to the static baseline. Moreover, our method even outperforms the large fixed-size VGG-11 by $0.20p.p$ test accuracy, while taking only $52.91\%$ of its training cost. For MobileNetV1, our method also achieves the best trade-off between training efficiency and test accuracy among all competitors.

**ResNet-18, VGG-19, and MobileNetV1 on CIFAR-100.** We also evaluate all methods on CIFAR-100 using different network architectures. Results in Table 3 show that Firely consistently achieves better test accuracy than Firefly-split, suggesting that adding new neurons provides more flexibility for exploration than merely splitting. Both Firely and our method achieve better performance than the original VGG-19, suggesting that network growing might have an additional regularizing effect. Our method yields the best accuracy and largest training cost reduction.

Table 4: ResNet-50 and MobileNetV1 on ImageNet.

| Method | ResNet-50 | | MobileNet-v1 | |
| | Train Cost(%) ↓ | Test Acc.(%) | Train Cost(%) ↓ | Test Acc.(%) |
| --- | --- | --- | --- | --- |
| Large | 100 | $76.72 \pm 0.18$ | 100 | $70.80 \pm 0.19$ |
| Net2Net | 60.12 | $74.89 \pm 0.21$ | 63.72 | $66.19 \pm 0.20$ |
| FireFly | 71.20 | $75.01 \pm 0.11$ | 86.67 | $66.40 \pm 0.14$ |
| GradMax | - | - | 86.67 | $68.60 \pm 0.20$ |
| Ours | **60.12** | $\mathbf{75.90 \pm 0.14}$ | **63.72** | $\mathbf{69.91 \pm 0.16}$ |

Table 5: Transformer on IWSLT'14.

| Method | Transformer | |
| | Train Cost(%) ↓ | BLEU↑ |
| --- | --- | --- |
| Large | 100 | $32.82 \pm 0.21$ |
| Net2Net | 64.64 | $30.97 \pm 0.35$ |
| Ours-w/o buffer | 64.64 | $31.44 \pm 0.18$ |
| Ours-w buffer | 64.64 | $31.62 \pm 0.16$ |
| Ours-w buffer-RA | 64.64 | $\mathbf{32.01 \pm 0.16}$ |

## 4.2 ImageNet Results

We first grow ResNet-50 on ImageNet and compare the performance of our method to Net2Net and FireFly under the same epoch schedule: $\{4, 4, 5, 6, 8, 9, 11, 14, 29\}$ (90 total) with 9 growing phases. We also grow MobileNetV1 from a small seed architecture, which is more challenging than ResNet-50. We train Net2Net and our method uses the same scheduler as for ResNet-50. We also compare with Firefly-Opt (a variant of FireFly) and GradMax and report their best results from [10]. Note that both methods not only adopt additional local optimization but also train with a less efficient growing scheduler: the final full-sized architecture needs to be trained for a $75\%$ fraction while ours only requires $32.2\%$. Table 4 shows that our method dominates all competing approaches.

## 4.3 IWSLT14 De-En Results

We grow a Transformer from $d_{model} = 64$ to $d_{model} = 512$ within 4 stages, each trained with 5 epochs using Adam. Applying gradient-based growing methods to the Transformer architecture is nontrivial due to their domain specific design of local optimization. As such, we only compare with Net2Net. We also design variants of our method for self-comparison, based on the adaptation rules for Adam in Appendix C. As shown in Table 5, our method generalizes well to the Transformer architecture.

## 4.4 Analysis

**Ablation Study.** We show the effects of turning on/off each of our modifications to the baseline optimization process of Net2Net (1) Growing: adding neurons with functionality preservation. (2) Growing+VT: only applies variance transfer. (3) Growing+RA: only applies LR rate adaptation. (4) Full method. We conduct experiments using both ResNet-20 on CIFAR-10 and ResNet-18 on CIFAR-100. As shown in Table 6, different variants of our growing method not only outperform Net2Net consistently but also reduce the randomness (std. over 3 runs) caused by random replication. We also see that, both RA and VT boost the baseline growing method. Both components are designed and woven to accomplish the empirical leap. Our full method bests the test accuracy.

| Variant | Res-20 on C-10 (%) | Res-18 on C-100 (%) |
|---------|--------------------|--------------------|
| Net2Net | $91.60 \pm 0.21 (+0.00)$ | $76.48 \pm 0.20 (+0.00)$ |
| Growing | $91.62 \pm 0.12 (+0.02)$ | $76.82 \pm 0.17 (+0.34)$ |
| Growing+VT | $92.00 \pm 0.10 (+0.40)$ | $77.27 \pm 0.14 (+0.79)$ |
| Growing+RA | $92.24 \pm 0.11 (+0.64)$ | $77.74 \pm 0.16 (+1.26)$ |
| Full | $92.53 \pm 0.11 (+0.93)$ | $78.12 \pm 0.15 (+1.64)$ |

Table 6: Ablation study on VT and RA components.

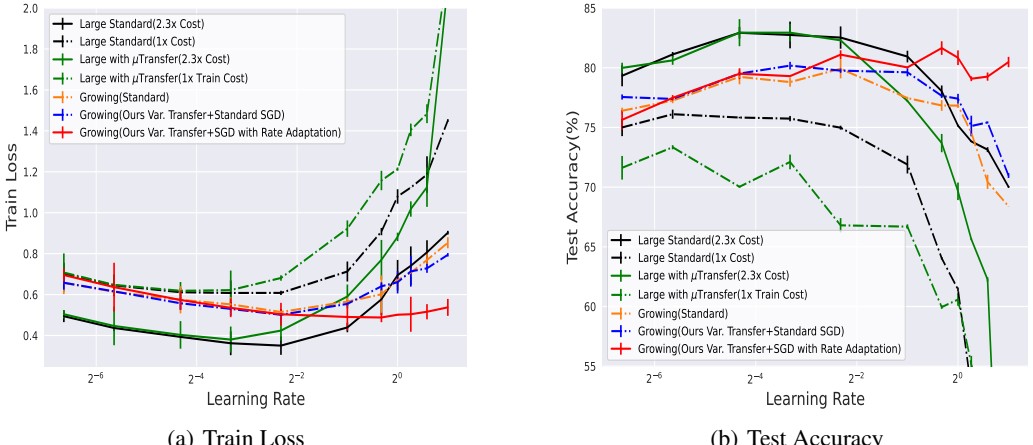

(a) Train Loss   (b) Test Accuracy

Figure 3: Baselines of 4-layer simple CNN.

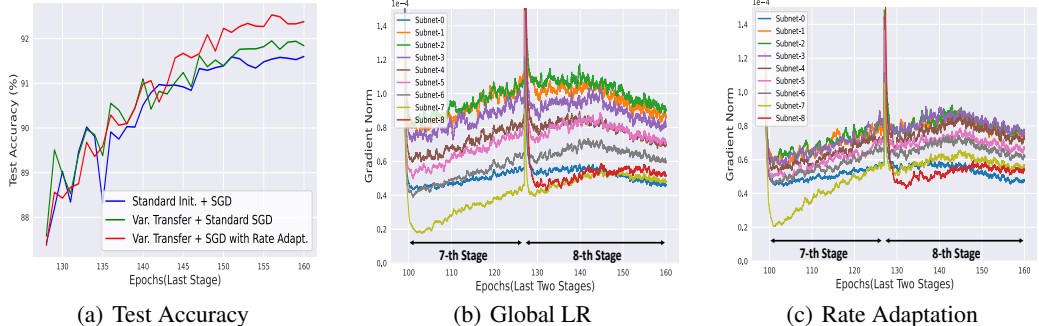

| (a) Test Accuracy | (b) Global LR | (c) Rate Adaptation |

Figure 4: (a) Performance with Var. Transfer and Rate Adaptation growing ResNet-20. (b) and (c) visualize the gradients for different sub-components along training in the last two stages.

**Justification for Variance Transfer.** We train a simple neural network with 4 convolutional layers on CIFAR-10. The network consists of 4 resolution-preserving convolutional layers; each convolution has 64, 128, 256 and 512 channels, a $3 \times 3$ kernel, and is followed by BatchNorm and ReLU activations. Max-pooling is applied to each layer for a resolution-downsampling of 4, 2, 2, and 2. These CNN layers are then followed by a linear layer for classification. We first alternate this network into four variants, given by combinations of training epochs $\in \{13(1\times), 30(2.3\times)\}$ and initialization methods $\in \{\text{standard}, \mu\text{transfer} [41]\}$. We also grow from a thin architecture within 3 stages, where the channel number of each layer starts with only 1/4 of the original one, *i.e.,* $\{16, 32, 64, 128\} \rightarrow \{32, 64, 128, 256\} \rightarrow \{64, 128, 256, 512\}$, each stage is trained for 10 epochs.

For network growing, we compare the baselines with standard initialization and variance transfer. We train all baselines using SGD, with weight decay set as 0 and learning rates sweeping over $\{0.01, 0.02, 0.05, 0.1, 0.2, 0.5, 0.8, 1.0, 1.2, 1.5, 2.0\}$. In Figure 3(b), growing with Var. Transfer (blue) achieves overall better test accuracy than standard initialization (orange). Large baselines with merely $\mu$transfer in initialization consistently underperform standard ones, which validate that the compensation from the LR rescaling is necessary in [41]. We also observe, in both Figure 3(a) and 3(b), all growing baselines outperform fixed-size ones under the same training cost, demonstrating positive regularization effects. We also show the effect of our initialization scheme by comparing test performance on standard ResNet-20 on CIFAR-10. As shown in Figure 4(a), compared with standard initialization, our variance transfer not only achieves better final test accuracy but also appears more stable. See Appendix F for a fully-connected network example.

**Justification for Learning Rate Adaptation.** We investigate the value of our proposed stage-wise learning rate adaptation as an optimizer for growing networks. As shown in the red curve in Figure 3, rate adaptation not only bests the train loss and test accuracy among all baselines, but also appears to be more robust over different learning rates. In Figure 4(a), rate adaptation further improves final test accuracy for ResNet-20 on CIFAR-10, under the same initialization scheme.

Figure 4(b) and 4(c) visualize the gradients of different sub-components for the 17-th convolutional layer of ResNet-20 during last two growing phases of standard SGD and rate adaptation, respectively. Our rate adaptation mechanism rebalances subcomponents' gradient contributions to appear in lower divergence than global LR, when components are added at different stages and trained for different durations. In Figure 5, we observe that the LR for newly added Subnet-8 (red) in last stage starts around $1.8\times$ the base LR, then quick adapts to a smoother level. This demonstrates that our method is able to automatically adapt the updates applied to new weights, without any additional local optimization costs [39, 10]. All above show our method has a positive effect in terms of stabilizing training dynamics, which is lost if one attempts to train different subcomponents using a global LR scheduler. Appendix D provides more analysis.

**Flexible Growing Scheduler.** Our growing scheduler gains the flexibility to explore the best trade-offs between training budgets and test performance in a unified configuration scheme (Eq. 6 and Eq. 7). We compare the exponential epoch scheduler ($p_t \in \{0.2, 0.25, 0.3, 0.35\}$) to a linear one ($p_t = 0$) in ResNet-20 growing on CIFAR-10, denoted as 'Exp.' and 'Linear' baselines in Figure 6. Both baselines use the architectural schedulers with $p_c \in \{0.2, 0.25, 0.3, 0.35\}$, each generates trade-offs between train costs and test accuracy by alternating $T_0$. The exponential scheduler yields better

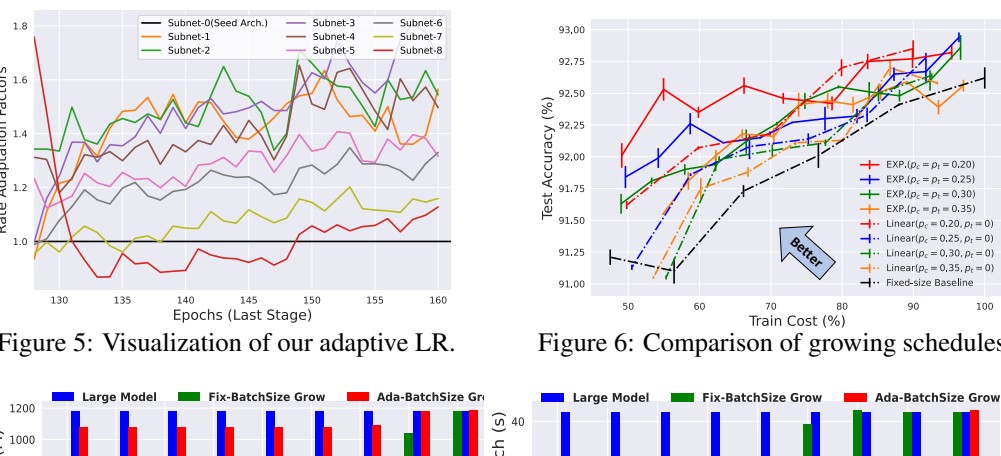

Figure 5: Visualization of our adaptive LR.

Figure 6: Comparison of growing schedules.

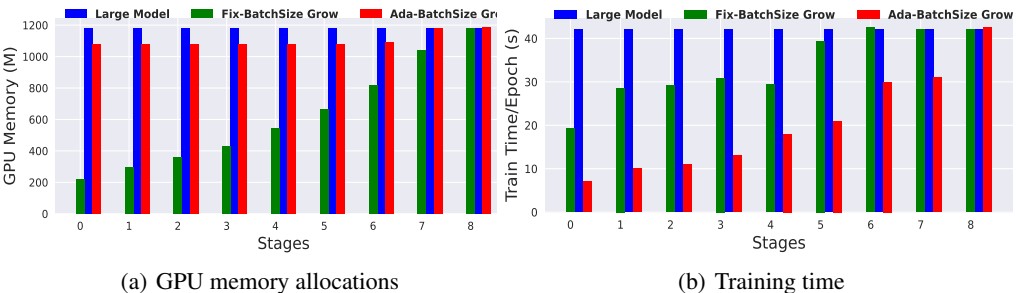

(a) GPU memory allocations

(b) Training time

Figure 7: Track of GPU memory and wall clock training time for each growing phase of ResNet-18.

overall trade-offs than the linear one with the same $p_c$. In addition to different growing schedulers, we also plot a baseline for fixed-size training with different models. Growing methods with both schedulers consistently outperform the fixed-size baselines, demonstrating that the regularization effect of network growth benefits generalization performance.

**Wall-clock Training Speedup.** We benchmark GPU memory consumption and wall-clock training time on CIFAR-100 for each stage during training on single NVIDIA 2080Ti GPU. The large baseline ResNet-18 trains for 140 minutes to achieve 78.36% accuracy. As shown in the green bar of Figure 7(b), the growing method only achieves marginal wall-clock acceleration, under the same fixed batch size. As such, the growing ResNet-18 takes 120 minutes to achieve 78.12% accuracy. The low GPU utilization in the green bar in Figure 7(a) hinders FLOPs savings from translating into real-world training acceleration. In contrast, the red bar of Figure 7 shows our batch size adaptation results in a large proportion of wall clock acceleration by filling the GPU memory, and corresponding parallel execution resources, while maintaining test accuracy. ResNet-18 trains for 84 minutes (1.7× speedup) and achieves 78.01% accuracy.

## 5 Conclusion

We tackle a set of optimization challenges in network growing and invent a corresponding set of techniques, including initialization with functionality preservation, variance transfer and learning rate adaptation to address these challenges. Each of these techniques can be viewed as 'upgrading' an original part for training static networks into a corresponding one that accounts for dynamic growing. There is a one-to-one mapping of these replacements and a guiding principle governing the formulation of each replacement. Together, they accelerate training without impairing model accuracy – a result that uniquely separates our approach from competitors. Applications to widely-used architectures on image classification and machine translation tasks demonstrate that our method bests the accuracy of competitors while saving considerable training cost.

## Acknowledgments and Disclosure of Funding

This work was supported by the National Science Foundation under grant CNS-1956180 and the University of Chicago CERES Center.

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

# A    More Analysis on Variance Transfer

Our fixed rescaling formulation, regarding relative network width, is an extension to the principled zero-shot HP transfer method [40, 41], based on the stability assumption, denoted as VT. A dynamic rescaling based on actual old weight values is an alternative plausible implementation choice, denoted as VT-constraint.

**Theorem A.1.** *Suppose the goal is to enforce the unit variance feature, then the scaling factor of an input layer with weights $\boldsymbol{W}^x$ with input shape $C^x$ is $\sqrt{\frac{1}{C^x \mathbb{V}[\boldsymbol{W}^x]}}$, while for a hidden layer with weights $\boldsymbol{W}^u$ and input shape $C^u$, it is $\sqrt{\frac{1}{C^u \mathbb{V}[\boldsymbol{W}^u]}}$.*

*Proof.* Consider a hidden layer that computes $\boldsymbol{u} = \boldsymbol{W}^x \boldsymbol{x}$ followed by another layer that computes $\boldsymbol{h} = \boldsymbol{W}^u \boldsymbol{u}$ (ignoring activations for simplicity). At a growth step, the first layer's outputs change from

$$\boldsymbol{u}_t = \boldsymbol{W}_t^x \boldsymbol{x} \tag{9}$$

to

$$\boldsymbol{u}_{t+1} = [\boldsymbol{u}_t{}', \boldsymbol{u}_t{}'', \boldsymbol{u}_t{}''] = [s^x \boldsymbol{W}_t^x \boldsymbol{x}, \boldsymbol{V}_t^x \boldsymbol{x}, \boldsymbol{V}_t^x \boldsymbol{x}]. \tag{10}$$

where $s^x$ denotes the scaling factor applied to $\boldsymbol{W}^x$. The second layer's outputs change from

$$\boldsymbol{h}_t = \boldsymbol{W}_t^u \boldsymbol{u}_t \tag{11}$$

to

$$\boldsymbol{h}_{t+1} = [\boldsymbol{h}_t{}', \boldsymbol{h}_t{}'', \boldsymbol{h}_t{}''] = [s^u \boldsymbol{W}_t^u \boldsymbol{u}_t{}', \boldsymbol{V}_t^u \boldsymbol{u}_{t+1}, \boldsymbol{V}_t^u \boldsymbol{u}_{t+1}]. \tag{12}$$

where $s^u$ denotes the scaling factor applied to $\boldsymbol{W}^u$.

The variance of the features after the growth step are:

$$\mathbb{V}[\boldsymbol{u}_t{}'] = (s^x)^2 C^x \mathbb{V}[\boldsymbol{W}_t^x] \tag{13}$$

$$\mathbb{V}[\boldsymbol{u}_t{}''] = C^x \mathbb{V}[\boldsymbol{V}_t^x] \tag{14}$$

$$\mathbb{V}[\boldsymbol{h}_t{}'] = (s^u)^2 C_t^u \mathbb{V}[\boldsymbol{W}_t^u] \mathbb{V}[\boldsymbol{u}_t{}'] \tag{15}$$

$$\mathbb{V}[\boldsymbol{h}_t{}''] = \mathbb{V}[\boldsymbol{V}_t^u](C_t^u \mathbb{V}[\boldsymbol{u}_t{}'] + (C_{t+1}^u - C_t^u) \mathbb{V}[\boldsymbol{u}_t{}'']) \tag{16}$$

Given the goal of enforcing unit-variance features for across all four vectors, we get:

$$s^x = \sqrt{\frac{1}{C^x \mathbb{V}[\boldsymbol{W}_t^x]}} \implies \mathbb{V}[\boldsymbol{u}_t{}'] = 1 \tag{17}$$

$$s^u = \sqrt{\frac{1}{C_t^u \mathbb{V}[\boldsymbol{W}_t^u]}} \implies \mathbb{V}[\boldsymbol{h}_t{}'] = \mathbb{V}[\boldsymbol{u}_t{}'] = 1 \tag{18}$$

$$\mathbb{V}[\boldsymbol{V}_t^x] = \frac{1}{C^x} \implies \mathbb{V}[\boldsymbol{u}_t{}''] = 1 \tag{19}$$

$$\mathbb{V}[\boldsymbol{V}_t^u] = \frac{1}{C_{t+1}^u} \implies \mathbb{V}[\boldsymbol{h}_t{}''] = \frac{1}{C_{t+1}^u}(C_t^u \mathbb{V}[\boldsymbol{u}_t{}'] + (C_{t+1}^u - C_t^u) \mathbb{V}[\boldsymbol{u}_t{}'']) = 1. \tag{20}$$

$\square$

This differs from the default VT formulation in Section 3.1, which corresponds to scaling factors of $s^x = 1$ and $s^u = \sqrt{\frac{C_t^u}{C_{t+1}^u}}$

We compare the default VT with VT-constraint by growing ResNet-20 on CIFAR-10. As shown in Table 7, both VT and VT-constraint outperform the standard baseline, which suggests standard initialization is a suboptimal design in network growing. We also note that involving the weight statistics is not better than our simpler design, which suggests that enforcing the old and the new features to have the same variance is not a good choice.

| Variants | Test Acc. (%) |
|---|---|
| Standard | $91.62 \pm 0.12$ |
| VT-constraint | $91.93 \pm 0.12$ |
| VT | $\boldsymbol{92.00 \pm 0.10}$ |

Table 7: Comparisons among standard initialization, VT-constraint (Theorem A.1) and the default VT (Section 3.1) for growing ResNet-20 on CIFAR-10.

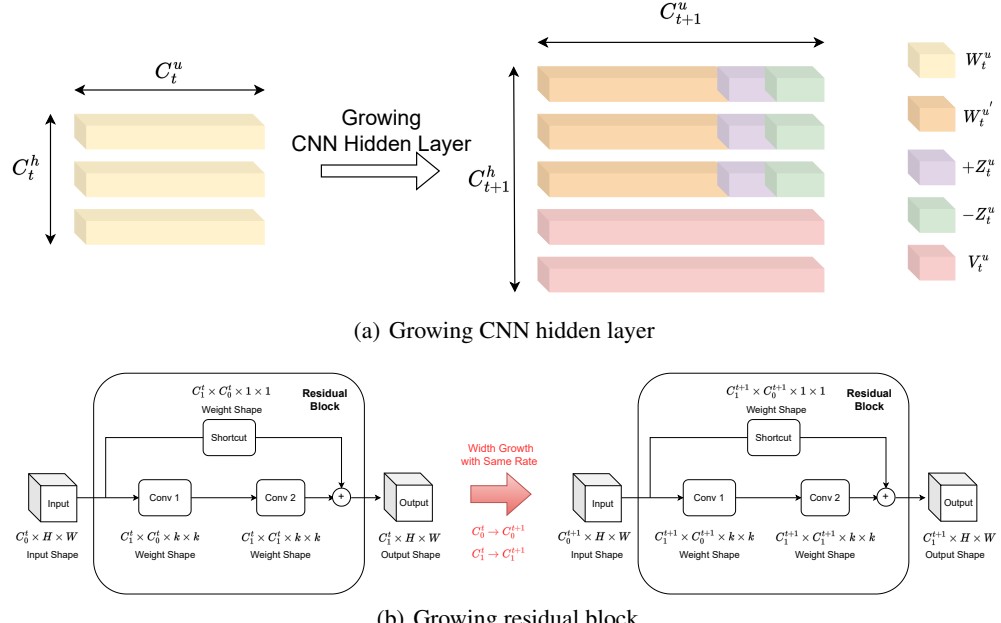

(a) Growing CNN hidden layer

(b) Growing residual block

Figure 8: Illustration for growing other layers.

# B  General Network Growing in Other Layers

We have shown network growing for 3-layer fully connected networks as a motivating example in Section 3.1. We now show how to generalize network growing with K (>3) layers, with conv layers, residual connections.

**Generalization to K ($> 3$) Layers.** In our network-width growing formulation, layers may be expanded in 3 patterns. (1) Input layer: output channels only; (2) Hidden layer: Both input (due to expansion of the previous layer) and output channels. (3) Output layer: input channels only. As such, the 3-layer case is sufficient to serve as a motivating example without loss of generality. For K ($> 3$) layer networks, the 2nd to $(K - 1)$th layers simply follow the hidden layer case defined in Section 3.1.

**Generalization to Convolutional layers.** In network width growth, we only need to consider the expansion along input and output channel dimensions no matter for fully connected networks or CNNs. Equations 1-4 still hold for CNN layers. Specifically, when generalizing to CNNs (from 2-d $C_{out} \times C_{in}$ weight matrices to 4-d $C_{out} \times C_{in} \times k \times k$ ones), we only consider the matrix operations on the first two dimensions since we do not change the kernel size $k$. For example, in Figure 2(b), newly added weights $+Z_t^u$ in linear layer can be indexed as $W[0 : C_t^h, C_t^u : C_t^u + \frac{C_{t+1}^u - C_t^u}{2}]$. In CNN layer, it is simply $W[0 : C_t^h, C_t^u : C_t^u + \frac{C_{t+1}^u - C_t^u}{2}, :, :]$.

**Residual Connections.** We note that differently from plain CNNs like VGG, networks with residual connections do require that the dimension of an activation and its residual match since these will be added together. Our method handles this by growing all layers at the same rate. Hence, at each growth step, the shape of the two tensors to be added by the skip connections always matches.

**Concrete Example of Adding Units to a Network.** In model width growth, our common practice is to first determine the output dimension expansion from the growth scheduler. If a previous layer's output channels are to be grown, then the input dimension of the current layer should be also expanded to accommodate that change.

Let's consider a concrete example for a CNN with more layers ($> 3$) and residual blocks. Without loss of generality, we omit the kernel size $k$ and denote each convolutional layer channel shape as $(C_{in}, C_{out})$. We denote the input feature with 1 output dimension as $x$. Initially, we have the input layer $W^1$ with channel dimensions of (1,2) to generate $h_1$ (2-dimensional), followed by a 2-layer residual block (identity mapping for residual connection) with weights $W^{(2)}(2, 2)$ and $W^{(3)}(2, 2)$, followed by an output layer with weights $W^{(4)}(2, 1)$.

The corresponding computation graph (omitting BN and ReLU for simplification) is

$$y = W^{(4)}\Big(W^{(3)}W^{(2)}W^{(1)}x + W^{(1)}x\Big),$$

In more detail, we rewrite the computation in the matrix formulation:

$$h^{(1)} = W^{(1)}x = \begin{bmatrix} w_{1,1}^{(1)} \\ w_{2,1}^{(1)} \end{bmatrix} x = \begin{bmatrix} h_1^{(1)} \\ h_2^{(1)} \end{bmatrix}$$

$$h^{(2)} = W^{(2)}h^{(1)} = \begin{bmatrix} w_{1,1}^{(2)} & w_{1,2}^{(2)} \\ w_{2,1}^{(2)} & w_{2,2}^{(2)} \end{bmatrix} \begin{bmatrix} h_1^{(1)} \\ h_2^{(1)} \end{bmatrix} = \begin{bmatrix} h_1^{(2)} \\ h_2^{(2)} \end{bmatrix}$$

$$h^{(3)} = W^{(3)}h^{(2)} = \begin{bmatrix} w_{1,1}^{(3)} & w_{1,2}^{(3)} \\ w_{2,1}^{(3)} & w_{2,2}^{(3)} \end{bmatrix} \begin{bmatrix} h_1^{(2)} \\ h_2^{(2)} \end{bmatrix} = \begin{bmatrix} h_1^{(3)} \\ h_2^{(3)} \end{bmatrix}$$

$$h^{(4)} = h^{(3)} + h^{(1)} = \begin{bmatrix} h_1^{(3)} \\ h_2^{(3)} \end{bmatrix} + \begin{bmatrix} h_1^{(1)} \\ h_2^{(1)} \end{bmatrix} = \begin{bmatrix} h_1^{(4)} \\ h_2^{(4)} \end{bmatrix}$$

$$y = W^{(4)}h^{(4)} = \begin{bmatrix} w_{1,1}^{(4)} & w_{1,2}^{(4)} \end{bmatrix} \begin{bmatrix} h_1^{(4)} \\ h_2^{(4)} \end{bmatrix}$$

Now assume that we want to grow the dimension of the network's hidden activations from 2 to 4 (*i.e.*, $h^{(1)}, h^{(2)}, h^{(3)}, h^{(4)}$, which are 2-dimensional, should become 4-dimensional each).

$$h^{(1)} = W^{(1)}x = \begin{bmatrix} w_{1,1}^{(1)} \\ w_{2,1}^{(1)} \\ w_{3,1}^{(1)} \\ w_{4,1}^{(1)} \end{bmatrix} x = \begin{bmatrix} h_1^{(1)} \\ h_2^{(1)} \\ h_3^{(1)} \\ h_4^{(1)} \end{bmatrix}$$

$$h^{(2)} = W^{(2)}h^{(1)} = \begin{bmatrix} w_{1,1}^{(2)} & w_{1,2}^{(2)} & w_{1,3}^{(2)} & w_{1,4}^{(2)} \\ w_{2,1}^{(2)} & w_{2,2}^{(2)} & w_{2,3}^{(2)} & w_{2,4}^{(2)} \\ w_{3,1}^{(2)} & w_{3,2}^{(2)} & w_{3,3}^{(2)} & w_{3,4}^{(2)} \\ w_{4,1}^{(2)} & w_{4,2}^{(2)} & w_{4,3}^{(2)} & w_{4,4}^{(2)} \end{bmatrix} \begin{bmatrix} h_1^{(1)} \\ h_2^{(1)} \\ h_3^{(1)} \\ h_4^{(1)} \end{bmatrix} = \begin{bmatrix} h_1^{(2)} \\ h_2^{(2)} \\ h_3^{(2)} \\ h_4^{(2)} \end{bmatrix}$$

$$h^{(3)} = W^{(3)}h^{(2)} = \begin{bmatrix} w_{1,1}^{(3)} & w_{1,2}^{(3)} & w_{1,3}^{(3)} & w_{1,4}^{(3)} \\ w_{2,1}^{(3)} & w_{2,2}^{(3)} & w_{2,3}^{(3)} & w_{2,4}^{(3)} \\ w_{3,1}^{(3)} & w_{3,2}^{(3)} & w_{3,3}^{(3)} & w_{3,4}^{(3)} \\ w_{4,1}^{(3)} & w_{4,2}^{(3)} & w_{4,3}^{(3)} & w_{4,4}^{(3)} \end{bmatrix} \begin{bmatrix} h_1^{(2)} \\ h_2^{(2)} \\ h_3^{(2)} \\ h_4^{(2)} \end{bmatrix} = \begin{bmatrix} h_1^{(3)} \\ h_2^{(3)} \\ h_3^{(3)} \\ h_4^{(3)} \end{bmatrix}$$

$$h^{(4)} = h^{(3)} + h^{(1)} = \begin{bmatrix} h_1^{(3)} \\ h_2^{(3)} \\ h_3^{(3)} \\ h_4^{(3)} \end{bmatrix} + \begin{bmatrix} h_1^{(1)} \\ h_2^{(1)} \\ h_3^{(1)} \\ h_4^{(1)} \end{bmatrix} = \begin{bmatrix} h_1^{(4)} \\ h_2^{(4)} \\ h_3^{(4)} \\ h_4^{(4)} \end{bmatrix}$$

$$y = W^{(4)}h^{(4)} = \begin{bmatrix} w_{1,1}^{(4)} & w_{1,2}^{(4)} & w_{1,3}^{(4)} & w_{1,4}^{(4)} \end{bmatrix} \begin{bmatrix} h_1^{(4)} \\ h_2^{(4)} \\ h_3^{(4)} \\ h_4^{(4)} \end{bmatrix}$$

We added 2 rows to the input layer's weights $W^{(1)}$ (to increase its output dimension from 2 to 4), added 2 rows and 2 columns to the hidden layer's weights $W^{(2)}, W^{(3)}$ (to increase its output dimensions from 2 to 4, and to increase its input dimension from 2 to 4 so they are consistent with the previous layers), and added 2 columns to the output layer's weights $W^{(4)}$ (to increase its input dimension from 2 to 4 so it is consistent with the increase

Table 8: Concrete growing example.

| Layer | Initial Arch | After Growth | New weights added |
|-------|-------------|--------------|-------------------|
| $W^{(1)}$ | (1,2) | (1,4) | $4 \times 1 - 2 \times 1 = 2$ |
| $W^{(2)}$ | (2,2) | (4,4) | $4 \times 4 - 2 \times 2 = 12$ |
| $W^{(3)}$ | (2,2) | (4,4) | $4 \times 4 - 2 \times 2 = 12$ |
| $W^{(4)}$ | (2,1) | (4,1) | $4 \times 1 - 2 \times 1 = 2$ |

in dimensionality of $h^{(4)}$). Note that $h^{(3)}$ and $h^{(1)}$ still maintain matching shapes (to be added up) in the residual block since we grow $W^{(1)}$ and $W^{(3)}$'s output dimensions with the same rate.

We summarize the architectural growth in terms of $C_{in}$ and $C_{out}$ (omitting kernel size $k$) in Table 8. We also show the growing for CNN hidden layer in Figure 8(a) and residual blocks in Figure 8(b) for better illustration.

Note that this example shows how growing works in general; our specific method also includes particulars as to what values are used to initialize the newly-added weights, as well as modifications to optimizer state.

## C  Generalization to Other Optimizers

We generalize our LR adaptation rule to Adam [20] and AvaGrad [31] in Table 9. Both methods are adaptive optimizers where different heuristics are adopted to derive a parameter-wise learning rate strategy, which provides primitives that can be extended using our stage-wise adaptation principle for network growing. For example, vanilla Adam adapts the global learning rate with running estimates of the first moment $\boldsymbol{m}_t$ and the second moment $\boldsymbol{v}_t$ of the gradients, where the number of global training steps $t$ is an integer when training a fixed-size model. When growing networks, our learning rate adaptation instead considers a vector $\boldsymbol{t}$ which tracks each subcomponent's 'age' (*i.e.,* number of steps it has been trained for). As such, for a newly grown subcomponent at a stage $i > 0$, $\boldsymbol{t}[i]$ starts as 0 and the learning rate is adapted from $\boldsymbol{m}_t / \sqrt{\boldsymbol{v}_t}$ (global) to $\frac{\boldsymbol{m}_{\boldsymbol{t}[i]} \setminus \boldsymbol{m}_{\boldsymbol{t}[i-1]}}{\sqrt{\boldsymbol{v}_{\boldsymbol{t}[i]} \setminus \boldsymbol{v}_{\boldsymbol{t}[i-1]}}}$ (stage-wise). Similarly, we also generalize our approach to AvaGrad by adopting $\boldsymbol{\eta}_t, d_t, \boldsymbol{m}_t$ of the original paper as a stage-wise variables.

**Preserving Optimizer State/Buffer.** Essential to adaptive methods are training-time statistics (*e.g.,* running averages $\boldsymbol{m}_t$ and $\boldsymbol{v}_t$ in Adam) which are stored as buffers and used to compute parameter-wise learning rates. Different from fixed-size models, parameter sets are expanded when growing networks, which in practice requires re-instantiating a new optimizer at each growth step. Given that our initialization scheme maintains functionality of the network, we are also able to preserve and inherit buffers from previous states, effectively maintaining the optimizer's state intact when adding new parameters. We experimentally investigate the effects of this state preservation.

Table 9: Rate adaptation rules for Adam [20] and AvaGrad [31].

| | | Our LR Adaptation |
|---|---|---|
| Adam | 0-th Stage | $\boldsymbol{m}_{\boldsymbol{t}[0]} / \sqrt{\boldsymbol{v}_{\boldsymbol{t}[0]}}$ |
| | i-th Stage | $\frac{\boldsymbol{m}_{\boldsymbol{t}[i]} \setminus \boldsymbol{m}_{\boldsymbol{t}[i-1]}}{\sqrt{\boldsymbol{v}_{\boldsymbol{t}[i]} \setminus \boldsymbol{v}_{\boldsymbol{t}[i-1]}}}$ |
| AvaGrad | 0-th Stage | $\frac{\boldsymbol{\eta}_{\boldsymbol{t}[0]}}{\|\boldsymbol{\eta}_{\boldsymbol{t}[0]} / \sqrt{d_{\boldsymbol{t}[0]}}\|_2} \odot \boldsymbol{m}_{\boldsymbol{t}[0]}$ |
| | i-th Stage | $\frac{\boldsymbol{\eta}_{\boldsymbol{t}[i]} \setminus \boldsymbol{\eta}_{\boldsymbol{t}[i-1]}}{\|\boldsymbol{\eta}_{\boldsymbol{t}[i]} \setminus \boldsymbol{\eta}_{\boldsymbol{t}[i-1]} / \sqrt{d_{\boldsymbol{t}[i]} - d_{\boldsymbol{t}[i-1]}}\|_2} \odot (\boldsymbol{m}_{\boldsymbol{t}[i]} \setminus \boldsymbol{m}_{\boldsymbol{t}[i-1]})$ |

Table 10: Generalization to Adam and AvaGrad for ResNet-20 on CIFAR-10.

| Optimizer | Training Method | Preserve Opt. Buffer | Train Cost (%) | Test Acc. (%) |
|---|---|---|---|---|
| Adam | Large fixed-size | NA | 100 | 92.29 |
| Adam | Growing | No | 54.90 | 91.44 |
| Adam | Growing | Yes | 54.90 | 91.61 |
| Adam+our RA. | Growing | Yes | 54.90 | **92.13** |
| AvaGrad | Large fixed-size | NA | 100 | 92.45 |
| AvaGrad | Growing | No | 54.90 | 90.71 |
| AvaGrad | Growing | Yes | 54.90 | 91.27 |
| AvaGrad+our RA. | Growing | Yes | 54.90 | **91.72** |

| Optimizer | Test Acc. (%) |
|---|---|
| Standard SGD | $91.95 \pm 0.09$ |
| SGD with Layer-wise Adapt. (LARS) | $91.32 \pm 0.11$ |
| Ours | $\mathbf{92.53 \pm 0.11}$ |

Table 11: Comparisons among standard SGD, LARS, and our adaptation method for growing ResNet-20 on CIFAR-10.

**Results with Adam and AvaGrad.** Table 10 shows the results of growing ResNet-20 on CIFAR-10 with Adam and Avagrad. For the large, fixed-size baseline, we train Adam with $lr = 0.1, \epsilon = 0.1$ and AvaGrad with $lr = 0.5, \epsilon = 10.0$, which yields the best results for ResNet-20 following [31]. We consider different settings for comparison: (1) optimizer without buffer preservation: the buffers are refreshed at each new growing phase; (2) optimizer with buffer preservation: the buffer/state is inherited from the previous phase, hence being preserved at growth steps; (3) optimizer with buffer and rate adaptation (RA): applies our rate adaptation strategy described in Table 9 while also preserving internal state/buffers. We observe that (1) consistently underperforms (2), which suggests that preserving the state/buffers in adaptive optimizers is crucial when growing networks. Option (3) bests the other settings for both Adam and AvaGrad, indicating that our rate adaptation strategy generalizes effectively to Adam and AvaGrad for the growing scenario. Together, these also demonstrate that our method has the flexibility to incorporate different statistics that are tracked and used by distinct optimizers, where we take Adam and AvaGrad as examples. Finally, our proposed stage-wise rate adaptation strategy can be employed with virtually any optimizer.

**Comparison with Layer-wise Adaptive Optimizer.** We also consider LARS [12, 43], a layer-wise adaptive variant of SGD, to compare different adaptation concepts: layer-wise versus layer + stage-wise (ours). Note that although LARS was originally designed for training with large batches, we adopt a batch size of 128 when growing ResNet-20 on CIFAR-10. We search the initial learning rate (LR) for LARS over {1e-3, 2e-3, 5e-3, 1e-2, 2e-2, 5e-2, 1e-1, 2e-1, 5e-1} and observe that a value of 0.02 yields the best results. We adopt the default initial learning rate of 0.1 for both standard SGD and our method. As shown in Table 11, LARS underperforms both standard SGD and our adaptation strategy, suggesting that layer-wise learning rate adaptation by itself – *i.e.,* without accounting for stage-wise discrepancies – is not sufficient for successful growing of networks.

# D  More Analysis on Rate Adaptation

We show additional plots of stage-wise rate adaptation when growing a ResNet-20 on CIFAR-10. Figure 9 shows the of adaptation factors based on the LR of the seed architecture from 1st to 8th stages (the stage index starts at 0). We see an overall trend that for newly-added weights, its learning rate starts at $> 1\times$ of the base LR then quickly adapts to a relatively stable level. This demonstrates that our approach is able to efficiently and automatically adapt new weights to gradually and smoothly fade in throughout the current stage's optimization procedure.

We also note that rate adaptation is a general design that different subnets should not share a global learning rate. The RA formulation is designed empirically. $max(1, ||\boldsymbol{W_i} \setminus \boldsymbol{W_{i-1}}||)$ is a plausible implementation choice,

| RA Implementation Choice | Test Acc. (%) |
|---|---|
| NA (Standard SGD) | $91.62 \pm 0.12$ |
| $max(1, ||\boldsymbol{W_i} \setminus \boldsymbol{W_{i-1}}||)$ | $91.42 \pm 0.12$ |
| Ours | $\mathbf{92.53 \pm 0.11}$ |

Table 12: Comparisons among different RA implementation choices for growing ResNet-20 on CIFAR-10.

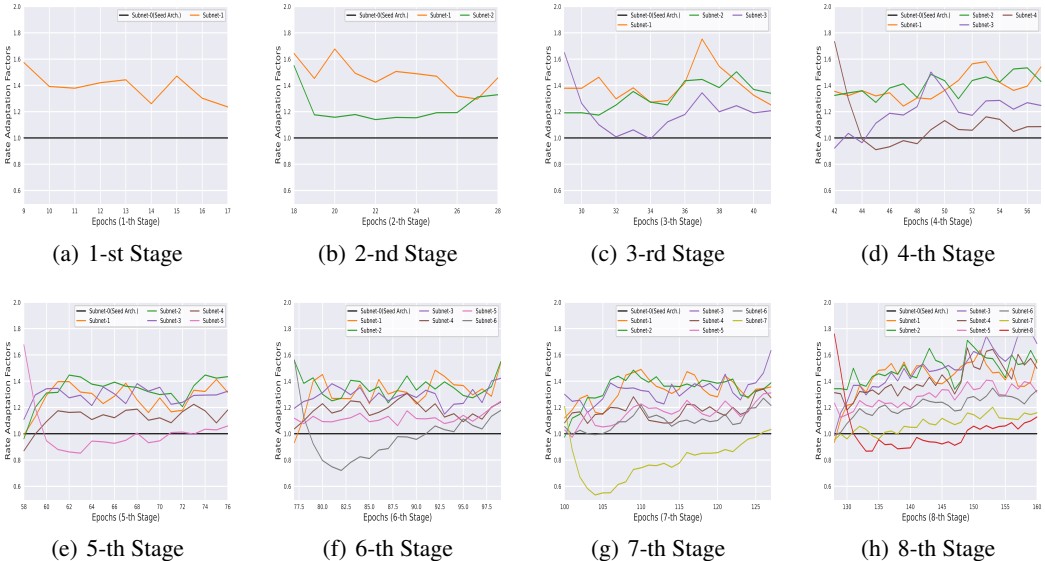

|  |  |  |  |
|---|---|---|---|
| (a) 1-st Stage | (b) 2-nd Stage | (c) 3-rd Stage | (d) 4-th Stage |
| (e) 5-th Stage | (f) 6-th Stage | (g) 7-th Stage | (h) 8-th Stage |

Figure 9: Visualization of rate adaptation factor dynamics across all growing stages (except 0-th)

based on the assumption that new weights must have a higher learning rate. We conducted experiments by growing ResNet-20 on CIFAR-10. As shown in Table 12, we see that this alternative does not work better than our original design, and even underperforms standard SGD.

# E   More Visualizations on Sub-Component Gradients

We further compare global LR and our rate adaptation by showing additional visualizations of sub-component gradients of different layers and stages when growing ResNet-20 on CIFAR-10. We select the 2nd (layer1-block1-conv1) and 17th (layer3-block2-conv2) convolutional layers and plot the gradients of each sub-component at the 3rd and 5th growing stages, respectively, in Figures 10, 11, 12, 13. These demonstrate that our rate adaptation strategy is able to re-balance and stabilize the gradient's contribution of different subcomponents, hence improving the training dynamics compared to a global scheduler.

# F   Simple Example on Fully-Connected Neural Networks

Additionally, we train a simple fully-connected neural network with 8 hidden layers on CIFAR-10 – each hidden layer has 500 neurons and is followed by ReLU activations. The network is has a final linear layer with 10 neurons for classification. Note that each CIFAR-10 image is flattened to a 3072-dimensional ($32 \times 32 \times 3$) vector as prior to being given as input to the network. We consider two variants of this baseline network by adopting training epochs (costs) $\in \{25(1\times), 50(2\times)\}$. We also grow from a thin architecture to the original one within 10 stages, each stage consisting of 5 epochs, where the number of units of each hidden layer grows from 50 to $100, 150, ..., 500$. The total training cost is equivalent to the fixed-size one trained for 25 epochs. We train all baselines using SGD, with weight decay set as 0 and learning rates sweeping over $\{0.01, 0.02, 0.05, 0.1, 0.2, 0.5\}$: results are shown in Figure 14(a). Compared to standard initialization (green),

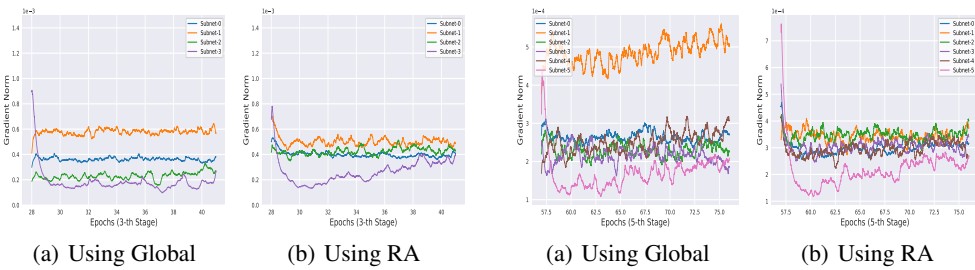

|  |  |  |  |
|---|---|---|---|
| (a) Using Global | (b) Using RA | (a) Using Global | (b) Using RA |

Figure 10: Gradients of 2nd conv at 3rd stage.   Figure 11: Gradients of 2nd conv at 5th stage.

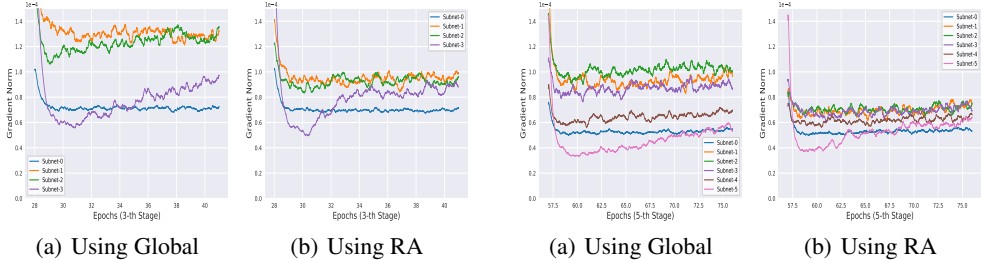

(a) Using Global      (b) Using RA         (a) Using Global      (b) Using RA

Figure 12: Gradients of 17th conv at 3rd stage.    Figure 13: Gradients of 17th conv at 5th stage.

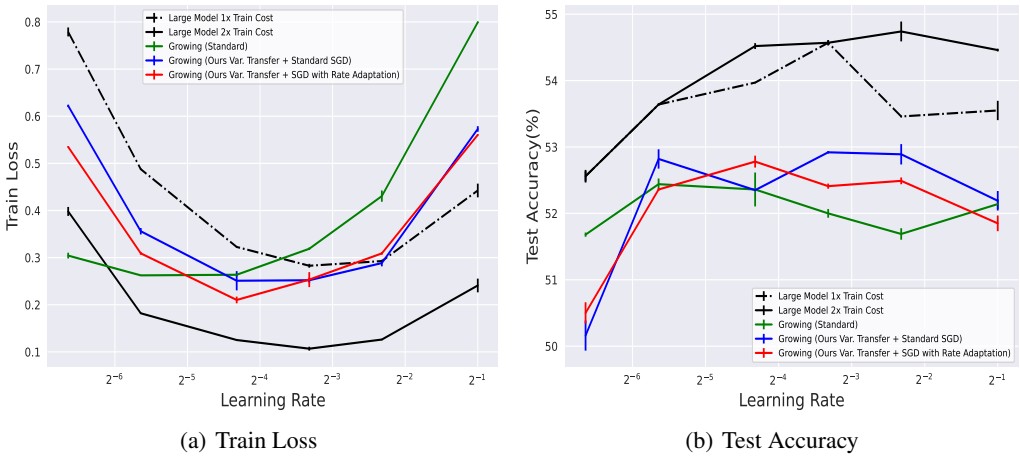

(a) Train Loss               (b) Test Accuracy

Figure 14: Results of simple fully-connected neural network.

the loss curve given by growing variance transfer (blue) is more similar to the curve of the large baseline – all using standard SGD – which is also consistent with the observations when training model of different scales separately [41]. Rate adaptation (in red) further lowers training loss. Interestingly, we observe in Figure 14(b) that the test accuracy behavior differs from the training loss one given in Figure 14(a), which may suggest that regularization is missing due to, for example, the lack of parameter-sharing schemes (like CNN) in this fully-connected network.

# G    More Comparisons with GradMax [10]

For CIFAR-10 and CIFAR-100, GradMax used different models for growing and we did not re-implement GradMax on both datasets. Also, generalizing such gradient-based growing methods to the Transformer architecture is nontrivial. As such, we only cover MobileNet on ImageNet which is used in both ours and GradMax. Our accuracy outperforms GradMax by 1.3 while lowering training costs, which is significant to demonstrate the benefit of our method. We also trained our method to grow WRN-28-1 (w/wo BatchNorm, used in GradMax paper) on CIFAR-10 and CIFAR-100 and compare it with GradMax in Table 13. We see that ours still consistently outperforms GradMax.

Table 13: Comparison with GradMax.

| Method | CIFAR-10 (w BN) | | CIFAR-10 (w/o BN) | | CIFAR-100 (w/o BN) | |
| | Train Cost (%) ↓ | Test Acc. (%) ↑ | Train Cost (%) ↓ | Test Acc. (%) ↑ | Train Cost (%) ↓ | Test Acc. (%) ↑ |
|---|---|---|---|---|---|---|
| Large Baseline | 100 | $93.40 \pm 0.10$ | 100 | $92.90 \pm 0.20$ | 100 | $69.30 \pm 0.10$ |
| GradMax [10] | 77.32 | $93.00 \pm 0.10$ | 77.32 | $92.40 \pm 0.10$ | 77.32 | $66.80 \pm 0.20$ |
| Ours | 58.24 | $93.29 \pm 0.12$ | 58.24 | $92.61 \pm 0.10$ | 58.24 | $67.83 \pm 0.15$ |

# H    Extension to Continuously Incremental Datastream

Another direct and intuitive application for our method is to fit continuously incremental datastream where $D_0 \subset D_1, ... \subset D_n... \subset D_{N-1}$. The network complexity scales up together with the data so that larger capacity can be trained on more data samples. Orthogonalized SGD (OSGD) [35] address the optimization difficulty in this context, which dynamically re-balances task-specific gradients via prioritizing the specific loss influence. We further extend our optimizer by introducing a dynamic variant of orthogonalized SGD, which progressively adjusts the priority of tasks on different subnets during network growth.

Suppose the data increases from $D_{n-1}$ to $D_n$, we first accumulate the old gradients $\boldsymbol{G_{n-1}}$ using one additional epoch on $D_{n-1}$ and then grow the network width. For each batch of $D_n$, we first project gradients of the new architecture ($n$-th stage), denoted as $\boldsymbol{G_n}$, onto the parameter subspace that is orthogonal to $\boldsymbol{G_{n-1}^{pad}}$, a zero-padded version of $\boldsymbol{G_{n-1}}$ with desirable shape. The final gradients $\boldsymbol{G_n^*}$ are then calculated by re-weighting the original $\boldsymbol{G_n}$ and its orthogonal counterparts:

$$\boldsymbol{G_n^*} = \boldsymbol{G_n} - \lambda * proj_{\boldsymbol{G_{n-1}^{pad}}}(\boldsymbol{G_n}), \quad \lambda : 1 \to 0 \tag{21}$$

where $\lambda$ is a dynamic hyperparameter which weights the original and orthogonal gradients. When $\lambda = 1$, subsequent outputs do not interfere with earlier directions of parameters updates. We then anneal $\lambda$ to 0 so that the newly-introduced data and subnetwork can smoothly fade in throughout the training procedure.

**Implementation Details.** We implement the task in two different settings, denoted as 'progressive class' and 'progressive data' on CIFAR-100 dataset within 9 stages. In the progressive class setting, we first randomly select 20 classes in the first stage and then add 10 new classes at each growing stage. In the progressive data setting, we sequentially sample a fraction of the data with replacement for each stage, *i.e.,* $20\%, 30\%, ..., 100\%$.

**ResNet-18 on Continuous CIFAR-100.** We evaluate our method on continuous datastreams by growing a ResNet-18 on CIFAR-100 and comparing the final test accuracies. As shown in Table 14, compared with the large baseline, our growing method achieves $1.53\times$ cost savings with a slight performance degradation in both settings. The dynamic OSGD variant outperforms the large baseline with $1.46\times$ acceleration, demonstrating that the new extension improves the optimization on continuous datastream through gradually re-balancing the task-specific gradients of dynamic networks.

Table 14: Growing ResNet-18 on incremental CIFAR-100.

| Method | Progressive Class | | Progressive Data | |
|---|---|---|---|---|
| | Train Cost (%) ↓ | Test Acc. (%) ↑ | Train Cost (%) ↓ | Test Acc. (%) ↑ |
| Large fixed-size Model | 100 | 76.80 | 100 | 76.65 |
| Ours | 65.36 | 76.50 | 65.36 | 76.34 |
| Ours-Dynamic-OSGD | 68.49 | 77.53 | 68.49 | 77.85 |

