# OpenReview forum: "Accelerated Training via Incrementally Growing Neural Networks using Variance Transfer and Learning Rate Adaptation"
_NeurIPS.cc/2023/Conference — NeurIPS 2023 poster_

### Official Review · Reviewer_8dnv · 2023-07-04

**Soundness:** 3 good
**Presentation:** 3 good
**Contribution:** 3 good
**Rating:** 7
**Confidence:** 4

**Summary:**

This paper introduces multiple synergistic tricks that allow for structurally growing networks to be competitive with and even sometimes outperform static networks. These methods include a batched schedule of layer widening, initializing opposing neurons, adapting the scale of the existing weights, handling optimizer  and using subnetwork-specific dynamic learning rates.

**Strengths:**

Originality: The paper introduces a novel synergy of methods to support growing a neural network, some of which are novel themselves.

Quality: The paper provides experimental results demonstrating that the proposed method achieves comparable or better accuracy than large fixed-size models while saving computation time during training across image classification and machine translation benchmarks and backbones architectures.

Clarity: Overall, the authors effectively communicate their ideas, methodologies, and results, making it easier for readers to understand the research. Some inconsistencies are noted below.

Significance: This is also the most thorough and practical work on growing models to my knowledge, from shallow and deep CNNs to transformers.


**Weaknesses:**

Please check and fix the formatting of your paper. Table 5 violates the right margin. Table 6 is hard to read without sufficient spacing between the columns. Many figures contain text that is too small and completely illegible when printed: for accessibility, please ensure all fonts within figures are legible at only 100% zoom.


**Questions:**

Figure 1(b), notably the input weights of $\delta+\epsilon_1$ and $\delta+\epsilon_2$, and the caption of Figure 2 suggest you are doing “symmetry breaking”, but the rest of your methods do not mention this and rather emphasize that the new weights are two identical copies. Can you clarify this point? Does symmetry breaking help empirically?

When including results of methods from related works, did you reimplement/run them yourself or copy their reported results? In the latter case, you must report any differences in experimental protocols beyond methods, including hyperparameters, optimizers, and software/hardware.

The learning rate adaptation method based on weight norm is not very clearly motivated. How was this selected: just on the previous works, or is there further justification? How is this affected by hyperparameter choices such as initialization scheme and weight decay? Other strategies exist that may be more intuitive, such as scaling by gradient norm [1]. Simply applying a subnet-wise cosine annealing may also suffice.

Lee, Yoonho, Annie S. Chen, Fahim Tajwar, Ananya Kumar, Huaxiu Yao, Percy Liang, and Chelsea Finn. "Surgical Fine-Tuning Improves Adaptation to Distribution Shifts." In The International Conference on Learning Representations. 2023.

Check the bibtex entry for Chen et al. (2016).

Whenever you refer to the appendix (Lines 116, 156, etc.), please refer to a specific appendix section.

Line 204 has a typo: CIAFR -> CIFAR.


**Limitations:**

No limitations nor potential negative societal impacts are explicitly addressed. Including these discussions would be appreciated.

---

> ### Author Rebuttal · Authors · 2023-08-10
>
> # To Reviewer 8dnv
>
> Thank you for the review and comments.  We address your points individually below.
>
> **Q: Formatting, margins, fonts.**
>
> A: In the final version, we will correct margins and spacing in Tables 5 and 6, and increase font size in figures.
>
> **Q: the caption of Figure 2 suggest you are doing “symmetry breaking”, but the rest of your methods do not mention this and rather emphasize that the new weights are two identical copies. Can you clarify this point? Does symmetry breaking help empirically?**
>
> A: We adopt symmetry breaking as suggested in Net2Net.  As indicated in Line 197, the noise for symmetry breaking is 0.001 to the norm of the initialization, which is negligible and does not affect our claim of functionality preservation during growth.  Symmetry breaking is necessary to enable the network to eventually learn to utilize all new parameters; otherwise identically initialized new parameters would receive exactly the same gradients and remain identical.
>
> **Q: When including results of methods from related works, did you reimplement/run them yourself or copy their reported results? In the latter case, you must report any differences in experimental protocols beyond methods, including hyperparameters, optimizers, and software/hardware.**
>
> A: We reimplemented all the methods except Gradmax-ImageNet and compare them under the same setting, including large fixed-size baseline architecture, hyperparamers, optimizers and software/hardware. For Gradmax on ImageNet, we use the reported results from the respective paper.  Our own trained large fixed-size MobileNet-V1 baseline yields the same performance (70.8) as that reported in the GradMax paper for the same architecture.  As we have consistent results between both codebases, our results are a fair comparison.  In Appendix Section 7, we also reimplement and compare with Gradmax on CIFAR-10 and CIFAR-100 under the same settings (and our method again outperforms Gradmax).
>
> **Q: No limitations nor potential negative societal impacts are explicitly addressed. Including these discussions would be appreciated.**
>
> A: We will add these discussions to the final version.
>
> **Q: The learning rate adaptation method based on weight norm is not very clearly motivated. How was this selected: just on the previous works, or is there further justification? How is this affected by hyperparameter choices such as initialization scheme and weight decay? Other strategies exist that may be more intuitive, such as scaling by gradient norm [1]. Simply applying a subnet-wise cosine annealing may also suffice.**
>
> A: Our learning rate adaptation (LRA) is a general design that different subnets should not share a global learning rate.  Our LRA formulation is designed empirically and we find scaling by weight norm works best for network growing with SGD.  Note that the LRA design choice is independent of the choice of other optimizer hyperparameters (HPs).  In our method, we choose the same set of HPs for a fair comparison.
>
> We do have some further investigation about the LRA design choices, as shown in Appendix Section 3 Table 5 with LARS (scaling by both weight norm and gradient norm) and Table 6 (new weights with higher learning rates).
>
> During the rebuttal, we also conducted experiments for ResNet-20 growth on CIFAR-10 using the other two LRA designs as suggested, with the same set of other optimizer HPs.  As shown in the table below, we find our method works best with SGD for network growing.
>
>  |LRA Impl. Choice|   Test Acc. (\%) $\uparrow$ |
> |--- |--- |
> |   NA (Standard SGD)	| $91.62\pm0.12$|
> |  Scaling Gradient	| $91.23\pm0.10$|
> |   Subnet-wise cosine annealing	| $91.98 \pm 0.11$|
> |  Ours | $92.53\pm0.11$|
>
>
> **Q: Check the bibtex entry for Chen et al. (2016); Refer to specific appendix section; Line 204 typo.**
>
> A: We will fix in the final version.

---

> > ### Comment · Reviewer_8dnv · 2023-08-14
> > **Response to rebuttal**
> >
> > I appreciate your response to each reviewer. I maintain my original rating.

---

### Official Review · Reviewer_NBgL · 2023-07-04

**Soundness:** 3 good
**Presentation:** 3 good
**Contribution:** 3 good
**Rating:** 7
**Confidence:** 4

**Summary:**

This paper studies the growth of neural networks in order to reduce training time. Authors study various optimization challenges when training dynamically grown neural networks and propose some novel techniques to address these challenges. The paper is well written in general, though it requires few clarifications.

**Strengths:**

- Important research topic with a lot of technical complexity. The community would benefit a lot from results and techniques given assuming authors plan to open-source their code and make their results reproducible.
- This paper studies the optimization for growing networks, which is an understudied, yet very important topic.
- This paper proposes a new initialization technique for growing NNs and combines it with a version of LARS optimizer.
- Experimental evaluation include transformers trained on natural language.
- Authors achieve real-time speed-ups through additional optimization (i.e. increasing batch size).


**Weaknesses:**

## Major
- Comparison with other methods in ImageNet is not ideal since baseline numbers might be different in different codebases. Also ideally one would only change the initialization and keep others parts the same to demonstrate that the proposed initialization is better than the alternatives. Looking at the results, it is not clear whether the alternatives would perform worse with the adaptive learning rate. For example one can run Gradmax with the proposed adaptive learning rate and growth schedule and compare the results. In general it would be nice to give more details about the Imagenet experiments and try applying the proposed improvements to other methods and see whether they also improve performance for them.
- “Together, they accelerate training without impairing model accuracy – a result that uniquely separates our approach from competitors.” This is not supported as baselines are not as good as the original baseline for the MobileNet-Cifar100 and Imagenet experiments. Can we train longer to get better results? It would be nice to run these experiments and check whether growing methods can match baseline performance with additional training.
- Contributions can be more accurate. (1) Function preserving growth is proposed by many (Openai, gradmax, net2net). (2) Stable training dynamics also exist for these methods. If what authors meant is to be more robust to different learning rates, I believe this is a common feature for adaptive optimizers.  (3) “Existing methods [26, 38] fail to derive a systemic way for distributing training resources across a growth trajectory.” Not clear this is true. Authors suggest a different heuristics than the ones used in previous work. I don't think there is enough motivation to claim this particular way being more systematic than the others. At the end they are all heuristics and the one proposed by this paper seem to perform better. I would focus on this part.

## Minor
- This paper seems to suggest a new initialization technique for neuron adding. You can also cite Gradmax showing that adding neurons to be more effective than splitting neurons.
- I would move VGG models to the appendix. For any practical purposes VGGs are a bit irrelevant at this point.
- Table 4: Firefly->FireflyOpt
- Figure 3 is very difficult to read in printed form, can you make these plots bigger?


**Questions:**

- Will the code be made public?
- How many epochs do you train for ImageNet?
- In Equation (5) what is f? Does it return momentum statistics? If so, it might be better to use a different letter instead of a function.
- Algorithm 1, what is S_n? Do you mean W_n?
- In Tables 2-5, how do you calculate the training cost? It would be nice to add this calculation to the appendix?
- In Tables 2-5, how do you generate Net2Net and Firefly results. Do you use the same schedule? If different schedules are used, do you do hyper parameter search for these baselines?
- Can you also report training loss in the appendix? Often small datasets have overfitting problem as shown in Figure 3, which is often not the case for larger datasets and it would be nice to know where the gains are.
- "We re-initialize the momentum buffer at each growing step when using SGD while preserving it for adaptive optimizers" Why optimizer states have different treatments in Adam and SGD? Can you do an ablation on this? Also do you do the same for the baselines, i.e. Net2Net and Firefly?

**Limitations:**

There is no limitation section at the moment. I think inability to match baseline numbers for larger experiments can be mentioned here.

---

> ### Author Rebuttal · Authors · 2023-08-10
>
> # To Reviewer NBgL
>
> Thank you for the review and comments.  We address your points individually below.
>
> **Q: Comparison with other methods in ImageNet is not ideal since baseline numbers might be different in different codebases; There is no limitation section at the moment. I think inability to match baseline numbers for larger experiments can be mentioned here.**
>
> A: For ImageNet, we reimplemented all the methods except Gradmax and compare them under the same setting, including large fixed-size baseline architecture, hyperparamers, optimizers and software/hardware.  For Gradmax on ImageNet, we use the reported results from the respective paper.  Our full-scale fixed-size MobileNet-V1 baseline is of the same performance (70.8) as reported in the Gradmax paper.  On the same network architectures, we achieve the same performance as existing codebases; our comparisons are fair.
>
> **Q: This is not supported as baselines are not as good as the original baseline for the MobileNet-Cifar100 and Imagenet experiments. Can we train longer to get better results? It would be nice to run these experiments and check whether growing methods can match baseline performance with additional training.**
>
> A: During the rebuttal, we trained MobileNet-Cifar100 longer and achieve equivalent performance while still saving on total training cost, as shown in the table below.
>
> |Method |Train Cost  (\%) $\downarrow$|  Test Acc. (\%) $\uparrow$ |
> |---  |--- |--- |
> |Large Baseline |   100 | $72.13\pm0.13$|
> |Ours |   52.90 | $71.53\pm0.13$|
> |Ours|   65.54 | $72.06\pm0.11$|
>
> We will add ImageNet results and adjust the discussion of the results accordingly in the final version.
>
> **Q: Contributions can be more accurate.**
>
> A: We will revise our writing to refer to specific technical methods and contributions rather than any overarching claim about high-level concepts.
>
> **Q: This paper seems to suggest a new initialization technique for neuron adding. You can also cite Gradmax showing that adding neurons to be more effective than splitting neurons.**
>
> A: We will add this in the final version.
>
> **Q: I would move VGG models to the appendix. For any practical purposes VGGs are a bit irrelevant at this point.**
>
> A: Thanks for the suggestion. We will move VGG experiments to Appendix.
>
> **Q: Table 4 typo; Figure 3.**
>
> A: We will fix the typo in Table 4; We will make Figure 3 bigger and move it to the appendix in the final version.
>
> **Q: Will the code be made public?**
>
> A: Yes.
>
> **Q: How many epochs do you train for ImageNet?**
>
> A: We follow the standard setting and train 90 epochs on ImageNet.
>
> **Q: In Equation (5) what is f? Does it return momentum statistics? If so, it might be better to use a different letter instead of a function.**
>
> A: $f$ denotes a general function
>
> **Q: Algorithm 1, what is $S\_n$? Do you mean $W\_n$?**
>
> A: Yes, it is $W\_n$. We will fix the typo in the revised version.
>
> **Q: In Tables 2-5, how do you calculate the training cost? It would be nice to add this calculation to the appendix?**
>
> A: We calculated the training costs by accumulating forward and backward FLOPs.  We then report the relative percentage to large fixed-size baselines, demonstrating the training efficiency.  We will add this explanation to the Appendix.
>
> **Q: In Tables 2-5, how do you generate Net2Net and Firefly results. Do you use the same schedule? If different schedules are used, do you do hyper parameter search for these baselines?**
>
> A: Yes, we use the same schedule for Net2Net and Firefly.
>
> **Q: Can you also report training loss in the appendix?**
>
> A: Thanks for the suggestion.  We will revise the appendix to include training loss.
>
> **Q: Why optimizer states have different treatments in Adam and SGD? Can you do an ablation on this? Also do you do the same for the baselines, i.e. Net2Net and Firefly?**
>
> A: In Appendix Section 3, 'Preserving Optimizer State/Buffer', we provide the justification for why we preserve the states during growing.  Appendix Table 4 provides the corresponding ablation study on Adam and Avagrad.
>
> For SGD, we find preserving the momentum buffer during growing does not improve the final performance. As such, we simply reinitialize it.  We did the same for all Net2Net and Firefly baselines.

---

> > ### Comment · Reviewer_NBgL · 2023-08-18
> > **Post Rebuttal**
> >
> > I read authors response. Authors addressed my concerns and provided some new results said more would be ready by camera ready. I updated my score accordingly.

---

### Official Review · Reviewer_GgUs · 2023-07-05

**Soundness:** 3 good
**Presentation:** 3 good
**Contribution:** 2 fair
**Rating:** 6
**Confidence:** 2

**Summary:**

Starting from an optimization and training dynamics perspective, this paper proposes a new method to efficiently growing neural networks, based on a parameterization that dynamically stabilizes weight, activation, and gradient scaling as the architecture grows over time and different subnetworks take the lion's share of the gradient contribution. To achieve this while preserving functional continuity, the approach  presented here grows the network by adding new random untrained neurons with initial net 0 functional contribution, and involves a learning rate scheduling mechanism that rebalances the gradient contribution to different subcomponents of the network added over time. The functional preservation reduces training disruptions and instabilities otherwise introduced by this step-function change in the network.This achieves wall-clock improvement at even or better accuracy than training a large fixed-size model.

**Strengths:**

The biggest strength of the paper is that it takes a reasonable, justified, principled approach to improving upon currently available methodologies in network growth, clearly hypothesizes that it might result in certain expected improvements, and then sets out to test this hypothesis through empirical experimentation. This thought process and methodology is clearly outlined in this well-written paper.

The paper compares to various competing baselines, not only to the large full-size model, thus providing an empirical comparison among similar solutions in the literature. In particular, the authors break down the performance of their method into incremental performance gains provided by the various components in the system they designed, as show in Table 6. These ablations help answer questions about the relative important of each design choice.

I also found the empirical investigation of the growing schedule quite compelling, as I imagine it would be a point of exploration for any practitioner wanting to implement this methodology.



**Weaknesses:**

The study is limited to simple, low scale image classification and machine translation tasks, which aren't representative of the application domains often facing researchers and practitioners these days. Therefore, it remains unproved whether this method would help in the context of large scale deep learning, which is the context in which the problems the authors are trying to solve actually occur. The disconnect between the problem statement and the regime in which the solution is tested weakens the point of the paper, as many promising approaches at low scales have proven completely ineffective at large scales.

One general concern with progressively grown architectures, especially with respect to whether the data distribution is / isn't changed with each growing iteration, is about the generalization properties of the final configuration, whether it loose the generalization properties of the equivalent full-scale architecture when the training data distribution stays constant, or whether it affects the task learning abilities in a continual learning setup.

**Questions:**

Just a suggestion: since Figure 3 is too small to be legible, I'd recommend moving it to the appendix.

**Limitations:**

There are no major concerns about societal impact of this work. Authors do not over-claim impact in this paper, and they are quite matter-of-fact about the applicability of their work. The scale limitation of their experiments, however, prevents this work from being more impactful.

---

> ### Author Rebuttal · Authors · 2023-08-10
>
> # To Reviewer GgUS
>
> Thank you for the review and comments.  We address your points individually below.
>
> **Q: The study is limited to simple, low-scale image classification and machine translation tasks, which aren't representative of the application domains often facing researchers and practitioners these days. ... The disconnect between the problem statement and the regime in which the solution is tested weakens the point of the paper, as many promising approaches at low scales have proven completely ineffective at large scales.**
>
> A: We demonstrate the proof-of-concept with results up to ResNet-50 on ImageNet scale.  We are contributing a new method, explaining the reasoning behind its design choices, and providing ablation studies to analyze its behavior at the largest scale our compute resources will allow.  We agree that extremely large-scale training (e.g., billion-scale parameter models) is a target for future work; we will need partners with access to that level of computing resources.
>
> **Q: One general concern with progressively grown architectures, especially with respect to whether the data distribution is / isn't changed with each growing iteration, is about the generalization properties of the final configuration, whether it loose the generalization properties of the equivalent full-scale architecture when the training data distribution stays constant, or whether it affects the task learning abilities in a continual learning setup.**
>
> A: Appendix Section 8 provides an extension to our approach under a continual learning setup.  Specifically, for the continuous data stream where data distribution may shift, we proposed a new Dynamic-OSGD method to accommodate the change along both data and architecture aspects.  In Appendix Table 8, we show that our growing method outperforms the equivalent full-scale architecture under the continual learning setting.  This is an initial step to bridging the dynamics of architectural configuration and data distribution during training.
>
> **Q: since Figure 3 is too small to be legible, I'd recommend moving it to the appendix.**
>
> A: We will move Figure 3 to the Appendix in the final version.

---

### Official Review · Reviewer_j2wb · 2023-07-07

**Soundness:** 3 good
**Presentation:** 3 good
**Contribution:** 3 good
**Rating:** 5
**Confidence:** 4

**Summary:**

The authors study the network growth problem and propose a novel problem to stabilize the optimization difficulty. The authors conduct experiments on various model architectures (including VGG, ResNet, MobileNet, and Transformer) and observe consistent performance improvements.

**Strengths:**

The studied problem is important. The proposed approach is novel. Experiments with various model architectures are conducted and demonstrate consistent performance improvements.

**Weaknesses:**

The most important application for the proposed algorithm is large-scale training. However, the experiments discussed in this study are based on relatively small models, which left doubts on the applicability of the proposed method on large-scale training.

**Questions:**

Is the Transformer used in Section 4.3 constructed in the Post-LN or Pre-LN manner?

**Limitations:**

I suggest conducting experiments with pre-training tasks like BERT, which would provide great value to this study. Also, as to the machine translation task, the IWSLT14 dataset is not as reliable as the WMT14 EN-De/Fr dataset. Some additional related work on this topic are: 1. Shallow-to-Deep Training for Neural Machine Translation; 2. On the transformer growth for progressive bert training.

---

> ### Author Rebuttal · Authors · 2023-08-10
>
> # To Reviewer j2wb
>
> Thank you for the review and comments.  We address your points individually below.
>
> **Q: The most important application for the proposed algorithm is large-scale training. However, the experiments discussed in this study are based on relatively small models, which left doubts on the applicability of the proposed method on large-scale training.**
>
> A: This paper demonstrates the proof-of-concept with results up to ImageNet scale.  We are contributing a new method, explaining the reasoning behind its design choices, and providing ablation studies to analyze its behavior at the largest scale our compute resources will allow.  We agree that extremely large-scale training (e.g., billion-scale parameter models) is a target for future work; we will need partners with access to that level of computing resources. We also appreciate the suggestions and will look into BERT pre-training and WMT14 for the final version.
>
> **Q. Is the Transformer used in Section 4.3 constructed in the Post-LN or Pre-LN manner?**
>
> A: It is constructed in the Pre-LN manner.  We will add this clarification to the final version.

---

> > ### Comment · Reviewer_j2wb · 2023-08-18
> > **Post Rebuttal**
> >
> > I read the author's response, and I maintain my original rating.

---

### Official Review · Reviewer_iBKk · 2023-07-13

**Soundness:** 2 fair
**Presentation:** 2 fair
**Contribution:** 2 fair
**Rating:** 5
**Confidence:** 4

**Summary:**

This paper developed an approach to efficiently grow neural network architecture by parameterization scheme with regard to weight, activation, and gradient scaling. It employed a learning rate adaptation mechanism rebalancing the gradient contribution of the subnetworks. The experimental results show efficiency in wall-clock training time.

**Strengths:**

-	The main technical section is described in detail, although there are some missing points (please refer to the Questions section.)
-	The paper is easy to follow.

**Weaknesses:**

-	Wondering if the authors considered the original training cost of the original backbone network. For example, if the original network costs XXX for training from scratch, how much will the proposed approach will cost when all steps’ costs are aggregated, apples to apples?
-	There is no description of how to calculate train costs.
-	It is not clear if the subnetworks are trained at a global learning rate or a different learning rate: at line 58, the authors claimed that subnetworks with different lengths have distinct learning rate schedules, while at line 142, it is noted that subnetworks from different growth stages will share a global learning rate.
-	It would have been better if the authors provided justification for why they picked the LR adaptation rule and comparisons with other LR adaptation rules. That is because a learning rate schedule (although, in many cases, it is overlooked) can have a high impact on the final performance and result. So it would have been nicer if the authors explained more or showed more clues on that.
-	In (6), C_0 (base case) is not defined.
-	In (7), the base case is not defined either. Also, are the parentheses (with floor opening and ceiling closing) for rounding to the nearest odd number?
-	It needs to be clarified why Figure 7 (b) is measured over time/epoch. In other words, why the time spent per epoch matters. It would have been nicer if the authors had added an explanation for that.
-	It is not clear the overall benefit of growing a network gradually while preserving the function computed by the model at each growth step. Is there any potential possibility/intention to use such an intermediate model? Otherwise, it is not clear to preserve the functions at each step.


**Questions:**

- In Figure 7 (b), why is it measured over time/epoch?
- In (6) and (7), what are the base cases?
- Why was the certain learning rate adaptation rule selected?

**Limitations:**

The authors did not mention anything about limitations. It would have been nicer if they were upfront about any limitations to share.

---

> ### Author Rebuttal · Authors · 2023-08-10
>
> # To Reviewer iBKk
>
> Thank you for the review.  However, it appears there is a serious misunderstanding as many of your questions make incorrect assumptions or are directly answered in the paper itself.  Please see our clarifications in response to your individiual points.
>
> **Q: Wondering if the authors considered the original training cost of the original backbone network. For example, if the original network costs XXX for training from scratch, how much will the proposed approach will cost when all steps’ costs are aggregated, apples to apples?**
>
> A: Training costs in Table 2-5 are apples-to-apples comparisons.  We train all methods from scratch and costs of all steps are aggregated in the comparison.  Growing methods do not require the training of an original backbone.  We do compare with the original backbone network training cost, denoted as Large Baseline, with 100\% costs in all tables.  For example, 54.90\% means the whole training process of network growing takes 54.90\% the cost of training a fixed-size full-scale architecture.
>
> **Q: There is no description of how to calculate train costs.**
>
> A: In Table 2-5, training costs are measured by accumulated forward and backward FLOPs.  We report the relative percentage to fixed-size baselines, demonstrating the training efficiency of network growing over full-scale architectures.  We will add this explanation to the revised version.
>
> **Q: It is not clear if the subnetworks are trained at a global learning rate or a different learning rate: at line 58, the authors claimed that subnetworks with different lengths have distinct learning rate schedules, while at line 142, it is noted that subnetworks from different growth stages will share a global learning rate.**
>
> A: In our method, subnetworks are trained with different learning rates.  Line 142 is not to describe our approach; instead, Line 142 points out that sharing a global learning rate may incur optimization difficulties.  Hence, we propose a learning rate adaptation approach to address this concern.
>
> **Q: It would have been better if the authors provided justification for why they picked the LR adaptation rule and comparisons with other LR adaptation rules.**
>
> A: Our learning rate adaptation (LRA) is a general design and one has the flexibility to design different rules empirically.  We choose the weight norm scaling rule which empirically works best with SGD.  As mentioned in Line 156, we do provide comparison to other LR adaptation rules in Appendix Tables 5 and 6 (see the supplementary material).
>
> **Q:  In (6), $C\_0$ (base case) is not defined; In (7), the base case is not defined either. Also, are the parentheses (with floor opening and ceiling closing) for rounding to the nearest odd number?**
>
> A: The base values vary for different baseline architectures.  We define them in the implementation details: Line 200 to 212.
>
> In Line 168, we define $\lfloor \cdot \rceil\_2$ as rounding to the nearest even number for Eq.(6). For Eq.(7), $\lfloor \cdot \rceil$ by default denotes rounding to the nearest integer.
>
> **Q: It needs to be clarified why Figure 7 (b) is measured over time/epoch.**
>
> A: In network growing, the network sizes, training epochs, and training costs vary in different stages.
> However, the network maintains the same size within each stage and time/epoch best represents the wall clock training efficiency for each stage.
>
> **Q: It is not clear the overall benefit of growing a network gradually while preserving the function computed by the model at each growth step. Is there any potential possibility/intention to use such an intermediate model? Otherwise, it is not clear to preserve the functions at each step.**
>
> A: Functionality preservation is a principle rule in all network growing methods, including Net2Net, Splitting, Firefly, and Gradmax.  This is to avoid the loss of information already learned from the previous training stages.  Disrupting functionality is equivalent to disrupting the learned behavior of the model, which means some additional training effort must be spent to relearn that behavior.  Adding new weights without functionality preservation leads to longer training time for equivalent performance.

---

> > ### Comment · Reviewer_iBKk · 2023-08-16
> >
> > I have increased my score to borderline accept.

---

### Official Review · Reviewer_Tyrx · 2023-07-27

**Soundness:** 3 good
**Presentation:** 3 good
**Contribution:** 3 good
**Rating:** 6
**Confidence:** 3

**Summary:**

This paper presents a method for incrementally growing network width across stages. The reader is shown the details of the growing process using a 3-layer feedforward network as an example, with the appendix providing details for other types of networks. Additional algorithmic details, including learning rate and batch size scheduling are provided. The paper then presents experimental results over 3 repeats, including an ablation study.

**Rebuttal:** I have read the rebuttal and it addresses my questions. Based on the novelty and contribution of this work, I maintain my original rating of 6.

**Strengths:**

Section 3 of this paper is perhaps the strongest part of this paper. There is no ambiguity in the presentation, and any questions the reader may have (generalization to Adam and other network architectures) are presented either in the appendix or later in the paper. The paper also does not limit itself to a single architecture, and shows details for two other optimizers, as well as other architecture types. It also provides an ablation study showing how each component of the overall approach is useful.

**Weaknesses:**

A few parts of the paper are a bit terse. In particular, Section 4 of the Appendix could benefit from additional exploration of, e.g., more alternatives or motivation/derivation for using the proposed alternative.

**Questions:**

- In Table 1, I'm not entirely sure I see a clear pattern: if we had one more layer, for example, would the last layer (whose dimensions I'll still call $C^h$) at initialization have the old weights rescaled as $\left(\frac{C^h_t}{C^h_{t+1}}\right)^{(3/2)}$, and the new inits be $\frac{1}{(C^h_{t+1})^3}$?
- In L143, you say using a global learning rate causes divergent behavior: have you seen this in practice? Can you show loss plots under this scheme?
- In Eq. 5, what is the motivation for setting $f$ to the norm function? Is the Appendix Section 4 what the paper refers to when discussing alternative heuristics (L156)?
- In Table 2, what do the underscores represent?
- In Tables 2-5, what statistical tests did you run to compare your methods? I'm also not sure 3 repeats is sufficient to capture variance in performance effectively.
- In Tables 4-5, please change the training costs of Net2Net to bold (since they were in bold in Tables 2-3).
- In Table 4, did Net2Net and Firefly achieve the same performance across all 3 runs, or did you only run each once?
- In Tables 2-5, how is training cost computed? Is it FLOPS/wall time/etc?

**Limitations:**

The paper's limitations are addressed by providing details for how to overcome them. For example, an ablation study shows the necessity of the learning rate adaptation scheme; the appendix provides details on expansion to residual blocks and convolutional blocks.

---

> ### Author Rebuttal · Authors · 2023-08-10
>
> # To Reviewer Tyrx
>
> Thank you for the review and comments. We address your points individually below.
>
> **Q: A few parts of the paper are a bit terse. In particular, Section 4 of the Appendix could benefit from additional exploration of, e.g., more alternatives or motivation/derivation for using the proposed alternative.**
>
> A: Thanks for the suggestion.  Since the learning rate adaption (LRA) formulation is designed empirically, one has the flexibility to implement other plausible options.  The motivation of the alternative design choice $max(||W\_{i} \setminus W\_{i-1}||)$ is to assign new weights with higher learning rates.  However, it does not work as well as baselines.
>
> **Q: In Table 1, I'm not entirely sure I see a clear pattern: if we had one more layer, for example, would the last layer (whose dimensions I'll still call $C\^h$), at initialization have the old weights rescaled as $(\frac{C\_t\^h}{C\_{t+1}\^{h}})\^{3/2}$, and the new inits be $\frac{1}{(C\_{t+1}\^{h})^2}$?**
>
> A: The patterns for initialization are as in Table 1, regardless of the number of layers.  For a network with K layers (K $\geq$ 3), the initialization rules still fall in three classes as depicted in Tabel 1: (1) 1st layer follows the 'Input Layer' pattern; (2) 2nd to (K-1)-th layers follow 'Hidden Layer' pattern; (3) K-th layer follows 'Output Layer' pattern.  Note that the initialization formulation in Table 1, regarding relative network width, is an extension to the principled zero-shot HP transfer method.  Table 1 takes a three-layer network as a motivating example and the patterns/rules generalize to arbitrary depth ($> 3$).
>
> **Q: you say using a global learning rate causes divergent behavior: have you seen this in practice? Can you show loss plots under this scheme?**
>
> A: We observed the divergent behavior of gradients caused by the global learning rate and plotted them in Figure 4(b) and also in Appendix Figures 5 and 6.  Note that, as shown in the ablation study in Table 6, even with a global learning rate, growing networks is still able to train (loss will not be divergent) but cannot achieve good performance.
>
> **Q: In Table 2, what do the underscores represent?**
>
> A: Underscores represents the second best results.  We will add this explanation to the Table 2 caption in the final version.
>
> **Q: In Tables 2-5, what statistical tests did you run to compare your methods? I'm also not sure 3 repeats is sufficient to capture variance in performance effectively.**
>
> A: We repeatedly train the models 3 times with different random seeds and then report the mean accuracy and standard deviation on the test set using these 3 models.  The observed standard deviations for results in Table 2-5 with 3 repeating runs are small, and imply that our method's performance improvements over competitors is statistically significant.
>
> **Q: In Tables 4-5, please change the training costs of Net2Net to bold (since they were in bold in Tables 2-3).**
>
> A: We will change it to bold in the final version.
>
> **Q: In Table 4, did Net2Net and Firefly achieve the same performance across all 3 runs, or did you only run each once?**
>
> A: For Net2Net and Firefly, we only run each once.  Note that our method's reported mean test accuracy and standard deviation establishes it as a significant improvement over these methods.  For completeness, we will repeat the ImageNet experiments with different random seeds for Net2Net and Firefly in the final version of the paper and include their standard deviations.
>
> **Q: In Tables 2-5, how is training cost computed? Is it FLOPS/wall time/etc?**
>
> A: In Table 2-5, the training cost is computed in FLOPS.  We also conducted a wall-clock time analysis in Figure 7.

---

### Author Rebuttal · Authors · 2023-08-10

# General Reply

We thank the reviewers for the detailed comments and answer specific questions in individual responses to each review below.  We also conducted additional experiments on learning rate adaption (prompted by Reviewer 8dnv) and slightly longer training (prompted by Reviewer NBgL).  Please see these results in the respective replies below.

---

### Decision · Program_Chairs · 2023-09-21

**Decision:**

Accept (poster)

**Comment:**

This paper introduces an efficient method for expanding neural network architectures through parameterization, encompassing weight, activation, and gradient scaling. It utilizes a learning rate adaptation mechanism to rebalance gradient contributions from subnetworks. Experimental results demonstrate notable reductions in wall-clock training time. All the reviewers unanimously accepted the paper, and so did the final decision. The authors should also revise their paper according to the reviewers' suggestions in the final version.